# Synthesis and analysis of 4-(3-fluoropropyl)-glutamic acid stereoisomers to determine the stereochemical purity of (4S)-4-(3-[18F]fluoropropyl)-L-glutamic acid ([18F]FSPG) for clinical use

**Kai-Ting Shih[1]◉, Ya-Yao Huang[2,3,4]◉, Chia-Ying Yang[1], Mei-Fang Cheng[2], Yu-Wen Tien[5], Chyng-Yann Shiue[2,3], Rouh-Fang Yen[2,3], Ling-Wei Hsin**[1,2,3,6]*

1 School of Pharmacy, College of Medicine, National Taiwan University, Taipei, Taiwan, 2 Department of Nuclear Medicine, National Taiwan University Hospital, Taipei, Taiwan, 3 Molecular Probes Development Core, Molecular Imaging Center, National Taiwan University, Taipei, Taiwan, 4 Institute of Medical Device and Imaging, College of Medicine, National Taiwan University, Taipei, Taiwan, 5 Department of Surgery, National Taiwan University Hospital, Taipei, Taiwan, 6 Center for Innovative Therapeutics Discovery, National Taiwan University, Taipei, Taiwan

◉ These authors contributed equally to this work.
* lwhsin@ntu.edu.tw

## Abstract

(4S)-4-(3-[18F]Fluoropropyl)-L-glutamic acid ([18F]FSPG) is a positron emission tomography (PET) imaging agent for measuring the system $x_C^-$ transporter activity. It has been used for the detection of various cancers and metastasis in clinical trials. [18F]FSPG is also a promising diagnostic tool for evaluation of multiple sclerosis, drug resistance in chemotherapy, inflammatory brain diseases, and infectious lesions. Due to the very short half-life (110 min) of 18F nuclide, [18F]FSPG needs to be produced on a daily basis; therefore, fast and efficient synthesis and analytical methods for quality control must be established to assure the quality and safety of [18F]FSPG for clinical use. To manufacture cGMP-compliant [18F]FSPG, all four nonradioactive stereoisomers of FSPG were prepared as reference standards for analysis. (2S,4S)-**1** and (2R,4R)-**1** were synthesized starting from protected L- and D-glutamate derivatives in three steps, whereas (2S,4R)-**1** and (2R,4S)-**1** were prepared in three steps from protected (S)- and (R)-pyroglutamates. A chiral HPLC method for simultaneous determination of four FSPG stereoisomers was developed by using a 3-cm Chirex 3126 column and a MeCN/CuSO4(aq) mobile phase. In this method, (2R,4S)-**1**, (2S,4S)-**1**, (2R,4R)-**1**, and (2S,4R)-**1** were eluted in sequence with sufficient resolution in less than 25 min without derivatization. Scale-up synthesis of intermediates for the production of [18F]FSPG in high optical purity was achieved via stereo-selective synthesis or resolution by recrystallization. The enantiomeric excess of intermediates was determined by HPLC using a Chiralcel OD column and monitored at 220 nm. The nonradioactive precursor with >98% ee can be readily distributed to other facilities for the production of [18F]FSPG. Based on the above accomplishments, cGMP-compliant [18F]FSPG met the acceptance criteria in specifications and

**Data Availability Statement:** All relevant data are within the manuscript and its Supporting information files.

**Funding:** This study was financially supported by the Ministry of Science and Technology of the Republic of China (Grant No. MOST 103-2320-B-002-012-MY3 and 105-2321-B-002-040). The funders had no role in study design, data collection and analysis, decision to publish, or preparation of the manuscript.

**Competing interests:** The authors have declared that no competing interests exist.

**Abbreviations:** FSPG, (4*S*)-4-(3-fluoropropyl)-L-glutamic acid; GMP, good manufacturing practices; PDAC, pancreatic ductal adenocarcinoma; PET, positron emission tomography.

was successfully manufactured for human use. It has been routinely prepared and used in several pancreatic ductal adenocarcinoma metastasis-related clinical trials.

## Introduction

Positron emission tomography (PET) is a noninvasive molecular imaging technique that uses special radiotracers and scanners to obtain cross-sectional images of the distribution of radiotracers in live animals and humans. From PET images, *in vivo* localization and quantification of brain and somatic proteins (e.g., transporters, receptors, and enzymes) and investigation of biochemical or physiologic processes (e.g., metabolic rate and blood flow) are feasible in both patients and healthy subjects. In addition, PET could be used to directly assess the effectiveness of novel agents at a very early stage, which enables earlier decision-making for selection of potential drug candidates and helps to elucidate the mechanisms of new drugs. Currently, PET and PET tracers have been widely used in oncology, cardiology, and neurology for diagnosis, in which [18F]fluorodeoxyglucose ([18F]FDG) is the most popular of the PET drugs used in clinics. Although [18F]FDG is useful in cancer diagnosis, staging, and treatment monitoring, new PET tracers aimed at other molecular targets for early tumor detection with higher sensitivity and specificity are still pursued by numerous researchers. Novel PET imaging agents could provide more specific pathological information other than the rate of glucose metabolism. This means they could furnish complementary data for the diagnosis and evaluation of therapy outcomes [1].

The system $x_C^-$ transporter (xCT) mediates sodium-independent cellular uptake of cystine in exchange for intracellular glutamic acid at the cell membrane and is highly expressed in the brain, pancreas, and immune cells. xCT has been reported to be expressed in hepatocellular carcinoma, breast cancer, and lung cancer [2] and plays a critical role in the regulation of the redox status in proliferating tumors [3–5]. Therefore, xCT is a promising target for PET molecular imaging of tumor metabolism and oxidative stress [6]. (4*S*)-4-(3-Fluoropropyl)-L-glutamic acid (FSPG, (2*S*,4*S*)-**1**) is a substrate of xCT and the 18F-labeled FSPG ([18F]FSPG) has been studied as a PET tracer for *in vivo* determination of xCT expression and activity (Fig 1) [6,7]. FSPG PET has been used for the detection of various cancers, such as brain [8], breast [7], lung, and liver cancers in clinical trials [8–12]. In addition, [18F]FSPG was found to be a more sensitive tool for the diagnosis and treatment evaluation of multiple sclerosis and detection of lung cancer with myocardial and pericardial metastases [13] in comparison to [18F]FDG [14,15]. FSPG PET could be used to measure the antioxidant capacity of tumor and predict drug resistance in chemotherapy [16–18]. Furthermore, [18F]FSPG is a potential tool for the diagnosis and therapy evaluation of inflammatory brain diseases (e.g., stroke) [19] and infectious lesions, as in sarcoidosis [20].

Early detection of liver metastases from pancreatic ductal adenocarcinoma (PDAC) is still an unmet medical need, which is a major reason for the low survival rate of PDAC patients. Contrast-enhanced computerized tomography (ceCT) and FDG PET are the most used methods to monitor metastases before surgery. However, due to the high uptake of [18F]FDG in the liver, the sensitivity, specificity and diagnostic accuracy of FDG PET for liver metastases from PDAC were not satisfactory. In an effort to develop a more sensitive diagnostic method for metastasized PDAC, we noticed that the xCT is highly expressed in the pancreas and PDAC, whereas the expression of xCT in the liver is significantly lower. Therefore, [18F]FSPG could be a more sensitive diagnostic drug for early detection of liver metastases from PDAC, which

**Fig 1. The structures of nonradioactive FSPG stereoisomers and [18F]FSPG.**

would provide valuable information for clinicians to choose more appropriate treatment strategies. It would also help to develop new drugs and therapies for metastasized PDAC and aforementioned diseases [21].

Recently, we found FSPG PET consistently found more PDAC metastases earlier than FDG PET in a rodent model [21]. Based on the promising preclinical data, we decided to conduct clinical trials to evaluate whether the FSPG PET is a more sensitive tool than FDG PET and ceCT in detecting liver metastases from PDAC. Due to the very short half-life of the 18F-labeled PET tracers (i.e., $t_{1/2}$ = 110 min), the preparation, purification, compounding, and quality control need to be completed in two hours to ready the cGMP-qualified [18F]FSPG injection for clinical use. However, [18F]FSPG has not been approved by any countries for routine clinical use yet. Therefore, the chemistry, manufacturing, and controls (CMC) for the production of [18F]FSPG need to be thoroughly studied to establish appropriate specifications of a qualified [18F]FSPG injection for clinical use. These are the information needed to submit an investigational new drug (IND) application to the local authority.

There are two chiral centers in FSPG, and only its (2S,4S)-**1** and (2S,4R)-**1** isomers have been reported [22]. In addition, no analytical method was yet available for detection and monitoring of the four FSPG stereoisomers. Here, we report 1) concise asymmetric synthesis to provide all four stereoisomers of FSPG (i.e., (2S,4S)-**1**, (2R,4R)-**1**, (2S,4R)-**1**, and (2R,4S)-**1**) as reference standards for analysis (Fig 1); 2) efficient analytical methods to monitor stereoisomeric purity of intermediates, four nonradioactive FSPG stereoisomers, and [18F]FSPG for quality control; 3) scale-up preparation of optically pure intermediates for preparation of [18F]FSPG. By integration of the above research accomplishments, the [18F]FSPG injection met the regulatory requirements and has been successfully manufactured for human use.

## Materials and methods

### General procedures and chemicals

1H- and 13C-NMR spectra were recorded on DPX-200, AV-400, and AV-600 FT-NMR spectrometers (Bruker, Germany) at 298 K. 1H-Decoupled 19F-NMR spectra were recorded at 377

MHz in $CD_3OD$ using an AVIII 400MHz NMR spectrometer (Bruker, Germany). Chemical shifts are expressed in parts per million (ppm) on the δ scale relative to a tetramethylsilane (TMS) internal standard. A solution of 5–20 mg of sample in 0.6–0.9 mL D-solvent was used in NMR experiments and no solvent signal suppression was used. Electron spray ionization (ESI) mass spectra and high-resolution mass measurements (HRMS) were obtained using an Esquire 2000 and Bruker Daltonik micro TOF mass spectrometer, respectively. Optical rotations were obtained using a Jasco P-2000 Digital Polarimeter (Tokyo, Japan) and are reported at the sodium D-line (589 nm). Thin-layer chromatography (TLC) was performed on Merck (art. 5715) silica gel plates and visualized under UV light (254 nm), upon treatment with iodine vapor, or upon heating after treatment with 5% phosphomolybdic acid in ethanol. Flash column chromatography was performed with Merck (art. 9385) 40–63 μm silical gel 60. Anhydrous tetrahydrofuran (THF) was distilled from sodium-benzophenone prior to use. No attempt was made to optimize yields.

Boc-L-Glu-OH was purchased from TCI-JP (Tokyo, Japan). Nitrophenyl sulfonyl chloride, (diethylamino)sulfur trifluoride (DAST), 4-(dimethylamino)pyridine (DMAP), and 1 M lithium bis(trimethylsilyl)amide (LiHMDS)/THF/ethylbenzene were purchased from Acros Organics (Geel, Belgium). Boc-D-Glu(OtBu)-OH, trifluoroacetic acid, and di-*tert*-butyl dicarbonate ($Boc_2O$) were purchased from Alfa Aesar (Lancashire, UK). 1-Bromo-3-fluoropropane, *N*-Boc-*D*-pyroglutamic acid ethyl ester, and *N*-Boc-*L*-pyroglutamic acid ethyl ester were purchased from AK Scientific (Union City, CA, USA). 3-Fluoropropanol was purchased from Matrix Scientific (Columbia, SC, USA) for the preparation of 3-fluoropropyl triflate. Dicyclohexyl-carbodiimide (DCC) and lithium hydroxide were purchased from Merck (Darmstadt, Germany). *N*-Boc-*L*-glutamic acid 5-*tert*-butyl ester, *tert*-butanol, and 1 M borane/THF complex were purchased from Sigma-Aldrich (St. Louis, MO, USA).

## Synthetic methods

**(4S)-4-(3-Fluoropropyl)-L-glutamic acid ((2S,4S)-1, FSPG) [6].** To a solution of (2*S*,4*S*)-**7** (85 mg, 0.20 mmol) in $CH_2Cl_2$ (1.5 mL) was added trifluoro acetic acid (TFA, 1.5 mL) dropwise. The mixture was stirred at room temperature for 1 h and then evaporated. The residue was triturated with diethyl ether to afford (2*S*,4*S*)-**1** (TFA salt, 50 mg, 0.16 mmol, 77%) as a white solid. $[\alpha]_D^{25}$ +15.6 (*c* 0.44, MeOH); $^1H$ NMR (400 MHz, methanol-$d_4$) δ 1.60–1.87 (m, 4H), 1.92–2.21 (m, 2H), 2.72–2.87 (m, 1H), 3.76–3.88 (m, 1H), 4.33–4.59 (m, 2H); $^{13}C$ NMR (100 MHz, methanol-$d_4$) δ 28.8 (d, *J* = 20.0 Hz), 29.3 (d, *J* = 5.0 Hz), 34.0, 42.6, 53.1, 84.4 (d, *J* = 163.0 Hz), 172.6, 178.0; $^{19}F$ NMR (377 MHz, methanol-$d_4$) δ −77.00 (TFA), −220.91; ESIMS *m/z* 208 ([M+H]$^+$); ESIHRMS calcd for $C_8H_{15}FNO_4$ [M+H]$^+$, 208.0985; found, 208.0967.

**(4R)-4-(3-Fluoropropyl)-D-glutamic acid ((2R,4R)-1).** A mixture of (2*R*,4*R*)-**7** (45 mg, 0.11 mmol), triisopropylsilane (42 μL), and $H_2O$ (42 μL) in TFA (1.6 mL), was stirred at room temperature for 2 h and then evaporated. The residue was triturated with diethyl ether, and then chromatographed on RediSep Rf Gold High Performance C18 (5.5 g; 20–40 μm) using 0.5% TFA in water to afford (2*R*,4*R*)-**1** (TFA salt, 14 mg, 0.04 mmol, 41%) as a white solid. $[\alpha]_D^{25}$ −16.0 (*c* 0.43, MeOH); $^1H$ NMR (400 MHz, methanol-$d_4$) δ 1.61–1.86 (m, 4H), 1.94–2.05 (m, 1H), 2.08–2.20 (m, 1H), 2.71–2.83 (m, 1H), 3.95 (dd, *J* = 8.7, 5.6 Hz, 1H), 4.33–4.43 (m, 1H), 4.45–4.55 (m, 1H); $^{13}C$ NMR (100 MHz, methanol-$d_4$) δ 28.8 (d, *J* = 20.0 Hz), 29.4 (d, *J* = 5.0 Hz), 33.6, 42.3, 52.4, 84.4 (d, *J* = 164.0 Hz), 171.6, 177.7; $^{19}F$ NMR (377 MHz, methanol-$d_4$) δ −77.00 (TFA), −220.91; ESIHRMS calcd for $C_8H_{15}NO_4F$ [M+H]$^+$, 208.0985; found, 208.0987.

**(2S,4R)-2-Amino-4-(3-fluoropropyl)pentanedioic acid ((2S,4R)-1).** A solution of (2$S$,4$R$)-**14** (22 mg, 0.072 mmol) in EtOAc and HCl$_{(conc)}$ (1/1, 1.0 mL) was stirred for 1 h and then evaporated. The residue was triturated with ether (2 mL × 3) to provide (2$S$,4$R$)-**1** (HCl salt, 17 mg, 0.070 mmol, 97%) as a colorless oil. $[\alpha]_D^{25}$ +11.2 ($c$ 0.58, MeOH); $^1$H NMR (400 MHz, D$_2$O) δ 1.63–1.79 (m, 4H), 1.94–1.99 (m, 1H), 2.28–2.35 (m, 1H), 2.66–2.71 (m, 1H), 3.97–4.01 (m, 1H), 4.40–4.53 (m, 2H); $^{13}$C NMR (100 MHz, D$_2$O) δ 26.9 (d, $J$ = 19.3 Hz), 27.8 (d, $J$ = 5.2 Hz), 31.6, 41.1, 51.7, 84.7 (d, $J$ = 157.6 Hz), 171.8, 178.5; $^{19}$F NMR (377 MHz, methanol-$d_4$) δ −220.80; ESIMS $m/z$ 208 [M+H]$^+$; HRESIMS calcd for C$_8$H$_{14}$FNNaO$_4$ [M+Na]$^+$, 230.0805; found, 230.0789.

**(2R,4S)-2-Amino-4-(3-fluoropropyl)pentanedioic acid ((2R,4S)-1).** To a solution of (2$R$,4$S$)-**14** (75 mg, 0.24 mmol) in CH$_2$Cl$_2$ (1.0 mL), 40% TFA in CH$_2$Cl$_2$ (1.0 mL) was added and stirred at room temperature for 1 h followed by evaporation. The residue was triturated with diethyl ether to afford (2$R$,4$S$)-**1** (TFA salt, 42 mg, 0.13 mmol, 54%) as a white solid. $[\alpha]_D^{25}$ −10.9 ($c$ 0.63, MeOH); $^1$H NMR (400 MHz, methanol-$d_4$) δ 1.60–1.87 (m, 4H), 1.87–1.98 (m, 1H), 2.35 (ddd, $J$ = 14.7, 9.2, 5.6 Hz, 1H), 2.61–2.76 (m, 1H), 3.97 (dd, $J$ = 8.1, 5.7 Hz, 1H), 4.38 (t, $J$ = 5.3 Hz, 1H), 4.50 (t, $J$ = 5.3 Hz, 1H); $^{13}$C NMR (100 MHz, methanol-$d_4$) δ 28.9 (d, $J$ = 19.9 Hz), 29.1 (d, $J$ = 5.1 Hz), 33.4, 42.1, 52.7, 84.4 (d, $J$ = 163.5 Hz), 171.8, 177.7; $^{19}$F NMR (377 MHz, methanol-$d_4$) δ −76.99 (TFA), −220.80; ESIHRMS calcd for C$_8$H$_{15}$NO$_4$F [M+H]$^+$, 208.0985; found, 208.0986.

**Di-tert-butyl (N-tert-butoxycarbonyl)-L-glutamate ((S)-3).** Di-$tert$-butyldicarbonate (1.66 g, 1.74 mL, 7.59 mmol), and DMAP (296 mg, 2.42 mmol) were added to a stirred solution of Boc-Glu(O$t$Bu)-OH ((S)-**4**, 2.45 g, 8.07 mmol) in $tert$-butanol (20 mL) at room temperature under N$_2$. After 30 min, the mixture was evaporated and the crude residue was chromatographed (silica gel, 5–10% EtOAc/$n$-hexane) to afford (S)-**3** (2.64 g, 7.34 mmol, 97%) as a white solid. $^1$H NMR (400 MHz, CDCl$_3$) δ 1.39–1.46 (m, 18H), 1.47 (s, 9H), 1.80–1.93 (m, 1H), 2.02–2.16 (m, 1H), 2.20–2.40 (m, 2H), 4.12–4.24 (m, 1H), 5.05 (d, $J$ = 7.8 Hz, 1H); $_{13}$C NMR (50 MHz, CDCl$_3$) δ 28.1, 28.2, 28.4, 31.7, 53.6, 79.8, 80.7, 82.1, 155.5, 171.7, 172.3; ESIHRMS calcd for C$_{18}$H$_{33}$NO$_6$Na [M+Na]$^+$, 382.2206; found, 382.2212.

**Di-tert-butyl (N-tert-butoxycarbonyl)-D-glutamate ((R)-3).** (R)-**3** was synthesized according to the procedure for the preparation of (S)-**3** using (R)-**4** to afford a white solid (81%). $^1$H NMR (400 MHz, CDCl$_3$) δ 1.39–1.54 (m, 27H), 1.79–1.97 (m, 1H), 1.99–2.19 (m, 1H), 2.19–2.44 (m, 2H), 4.08–4.32 (m, 1H), 5.06 (d, $J$ = 7.9 Hz, 1H); $_{13}$C NMR (100 MHz, CDCl$_3$) δ 28.1, 28.2, 28.4, 31.8, 53.7, 79.8, 80.7, 82.1, 155.5, 171.7, 172.3; ESIHRMS calcd for C$_{18}$H$_{33}$NO$_6$Na [M+Na]$^+$, 382.2206; found, 382.2209.

**Di-tert-butyl (2S,4S)-4-allyl-2-tert-butoxycarbonylamino-pentanedioate ((2S,4S)-5).** To a solution of (S)-**3** (200 mg, 0.56 mmol) in THF (2.8 mL), a solution of lithium bis(trimethylsilyl)amide in THF (1.0 M, 1.22 mL, 1.22 mmol) was added dropwise at -78˚C, and stirred for another 2 h. Allyl bromide (202 mg, 144 µL, 1.67 mmol) was then added dropwise, stirred for 1.5 h, and treated with 2 N HCl (2.8 mL) and EtOAc (9 mL). The organic layer was washed with water, dried over MgSO$_4$, filtered, and evaporated. The residue was purified by flash column chromatography (silica gel, 5–10% EtOAc/$n$-hexane) to afford (2$S$,4$S$)-**5** as a white solid (201 mg, 0.50 mmol, 91%). $^1$H NMR (400 MHz, CDCl$_3$) δ 1.40–1.50 (m, 27H), 1.79–1.95 (m, 2H), 2.24–2.39 (m, 2H), 2.43 (p, $J$ = 6.6 Hz, 1H), 4.14 (dd, $J$ = 15.4, 8.6 Hz, 1H), 4.90 (d, $J$ = 8.9 Hz, 1H), 4.99–5.18 (m, 2H), 5.63–5.82 (m, 1H); $_{13}$C NMR (100 MHz, CDCl$_3$) δ 28.1, 28.2, 28.4, 33.5, 36.8, 43.1, 53.2, 79.8, 80.9, 81.9, 117.5, 134.8, 155.6, 172.0, 174.5; ESIHRMS calcd for C$_{21}$H$_{37}$NO$_6$Na [M+Na]$^+$, 422.2519; found, 422.2513.

**Di-tert-butyl (2S,4S)-2-tert-butoxycarbonylamino-4-(3-hydroxypropyl)pentanedioate ((2S,4S)-6).** To a solution of (2$S$,4$S$)-**5** (1.00 g, 2.50 mmol) in THF (25 mL), BH$_3$•THF in THF (1.0 M, 3.0 mL, 3.00 mmol) was added dropwise at 0˚C under N$_2$, and stirred for 1 h.

NaOH$_{(aq)}$ (1.0 N, 3.25 mL) and H$_2$O$_2$ (35%, 2.79 mL) were added dropwise to the mixture and stirred for 30 min. The solution was diluted with water (50 mL), evaporated to remove THF, and then extracted with EtOAc (100 mL). The organic layer was washed with water (100 mL) and brine (100 mL×2), dried over MgSO$_4$, filtered, and evaporated. The crude residue was chromatographed (silica gel, 33% EtOAc/$n$-hexane) to afford (2$S$,4$S$)-**6** (884 mg, 2.12 mmol, 85%) as a white solid. $^1$H NMR (400 MHz, CDCl$_3$) δ 1.44 (s, 9H), 1.456 (s, 9H), 1.463 (s, 9H), 1.49–1.57 (m, 1H), 1.57–1.65 (m, 2H), 1.72–1.93 (m, 3H), 2.16 (t, $J$ = 5.5 Hz, 1H), 2.28–2.44 (m, 1H), 3.54–3.72 (m, 2H), 4.19 (td, $J$ = 9.2, 5.4 Hz, 1H), 5.01 (d, $J$ = 8.7 Hz, 1H); $^{13}$C NMR (50 MHz, CDCl$_3$) δ 27.6, 28.1, 28.2, 28.4, 29.9, 35.5, 42.3, 52.8, 61.4, 80.2, 80.9, 82.2, 156.0, 171.9, 175.2; ESIMS $m/z$ 440 ([M+Na]$^+$); ESIHRMS calcd for C$_{21}$H$_{39}$NO$_7$Na [M+Na]$^+$, 440.2624; found, 440.2640.

**Di-tert-butyl (2S,4S)-2-((tert-butoxycarbonyl)amino)-4-(3-fluoropropyl)pentanedioate ((2S,4S)-7).** Method A: To a solution of DAST (77 mg, 63 μL, 0.48 mmol) in CH$_2$Cl$_2$ (1.0 mL), a mixture of (2$S$,4$S$)-**6** (100 mg, 0.24 mmol) and diisopropylethylamine (DIPEA, 93 mg, 125 μL, 0.72 mmol) in CH$_2$Cl$_2$ (1.0 mL) was added dropwise at -78˚C, and then warmed to room temperature and stirred for another 3 days. The resulting solution was treated with NaHCO$_{3(sat)}$ (10 mL) and CH$_2$Cl$_2$ (10 mL). The organic layer was washed with brine, dried over MgSO$_4$, filtered, and evaporated. The crude residue was chromatographed (silica gel, 5–10% EtOAc/$n$-hexane) to afford (2$S$,4$S$)-**7** (49 mg, 0.12 mmol, 49%) as a pale yellow solid.

Method B: To a solution of ($S$)-**3** (500 mg, 1.29 mmol) in THF (7 mL) was added lithium bis(trimethylsilyl)amide in THF (1.0 M, 3.10 mL, 3.10 mmol) dropwise at -78˚C, and the mixture was stirred for another 15 min. 1-Bromo-3-fluoropropane (300 mg, 195 μL, 2.13 mmol) in THF (5 mL) was then added dropwise to the mixture, and stirred for 5 h at -78˚C. NH$_4$Cl$_{(sat)}$ (12.5 mL) was added and the resulting mixture was treated with CH$_2$Cl$_2$ (25 mL) and H$_2$O (12.5 mL). The organic extract was washed with brine, dried over MgSO$_4$, filtered, and evaporated. The crude product was chromatographed (silica gel, 5–10% EtOAc/$n$-hexane) to afford (2$S$,4$S$)-**7** (470 mg, 1.12 mmol, 80%) as a white solid. $^1$H NMR (400 MHz, CDCl$_3$) δ 1.43 (s, 9H), 1.46 (s, 18H), 1.63–1.77 (m, 4H), 1.78–1.98 (m, 2H), 2.30–2.45 (m, 1H), 4.14 (td, $J$ = 9.3, 5.0 Hz, 1H), 4.33–4.44 (m, 1H), 4.45–4.55 (m, 1H), 4.89 (d, $J$ = 8.8 Hz, 1H); $^{13}$C NMR (100 MHz, CDCl$_3$) δ 27.9 (d, $J$ = 20.0 Hz), 28.1, 28.2, 28.3 (d, $J$ = 5.0 Hz), 28.4, 34.5, 43.2, 53.3, 79.9, 81.0, 82.1, 83.7 (d, $J$ = 164.0 Hz), 155.7, 171.9, 174.8; ESIMS $m/z$ 442 ([M+Na]$^+$); ESIHRMS calcd for C$_{21}$H$_{38}$FNO$_6$Na [M+Na]$^+$, 442.2581; found, 442.2597.

**Di-tert-butyl (2R,4R)-2-((tert-butoxycarbonyl)amino)-4-(3-fluoropropyl)pentanedioate ((2R,4R)-7).** (2$R$,4$R$)-**7** was synthesized according to the procedure for the preparation of (2$S$,4$S$)-**7** (Method B) using ($R$)-**3** to afford a white solid (70%). $^1$H NMR (400 MHz, CDCl$_3$) δ 1.43 (s, 9H), 1.46 (s, 18H), 1.58–1.78 (m, 4H), 1.78–2.00 (m, 2H), 2.29–2.47 (m, 1H), 4.06–4.22 (m, 1H), 4.32–4.45 (m, 1H), 4.45–4.55 (m, 1H), 4.89 (d, $J$ = 8.9 Hz, 1H); $^{13}$C NMR (100 MHz, CDCl$_3$) δ 27.9 ($J$ = 20.0 Hz), 28.1, 28.2, 28.3 ($J$ = 6.0 Hz), 28.4, 34.5, 43.2, 53.3, 79.9, 81.0, 82.1, 83.7 ($J$ = 164.0 Hz), 155.7, 171.9, 174.8.; ESIHRMS calcd for C$_{21}$H$_{38}$NO$_6$FNa [M+Na]$^+$, 442.2581; found, 442.2583.

**(2S,4R)-1-tert-Butyl 2-ethyl 4-allyl-5-oxopyrrolidine-1,2-dicarboxylate ((2S,4R)-10) and (2S,4S)-1-tert-butyl 2-ethyl 4-allyl-5-oxopyrrolidine-1,2-dicarboxylate ((2S,4S)-10)[17].** To a solution of ($S$)-1-$tert$-butyl 2-ethyl 5-oxopyrrolidine-1,2-dicarboxylate (($S$)-**8**, 1.00 g, 3.89 mmol) in THF (19 mL), a solution of lithium bis(trimethylsilyl)amide in THF/ethylbenzene (1.0 M, 4.3 mL, 4.30 mmol) was added dropwise at -78˚C under N$_2$. The mixture was stirred for 1 h, and allyl bromide (1.35 mL, 15.6 mmol) in THF (13 mL) was added dropwise at -78˚C and stirred for 2 h. The resulting mixture was quenched with NH$_4$Cl$_{(sat.)}$ (100 mL), and extracted with CH$_2$Cl$_2$ (100 mL). The organic extract was washed with brine, dried over MgSO$_4$, filtered and evaporated. The residue was chromatographed (silica gel, 20~50%

EtOAc/$n$-hexane) to provide (2$S$,4$R$)-**10** (431 mg, 1.08 mmol, 37%) and (2$S$,4$S$)-**10** (203 mg, 0.47 mmol, 18%) as colorless oils. (2$S$,4$R$)-**10**: $^1$H NMR (600 MHz, CDCl$_3$) δ 1.29 (t, $J$ = 7.1 Hz, 3H), 1.50 (s, 9H), 1.98–2.04 (m, 1H), 2.16–2.22 (m, 2H), 2.61–2.65 (m, 1H), 2.71–2.76 (m, 1H), 4.23 (q, $J$ = 7.1 Hz, 2H), 4.54 (d, $J$ = 9.7 Hz,1H), 5.07–5.12 (m, 2H), 5.71–5.77 (m, 1H); $^{13}$C NMR (150 MHz, CDCl$_3$) δ 14.3, 27.9, 28.0, 34.6, 41.3, 57.3, 61.8, 83.7, 117.9, 134.5, 149.6, 171.4, 174.5. (2$S$,4$S$)-**10**: $^1$H NMR (600 MHz, CDCl$_3$) δ 1.30 (t, $J$ = 7.1 Hz, 3H), 1.50 (s, 9H), 1.70–1.75 (m, 1H), 2.19–2.25 (m, 1H), 2.45–2.50 (m, 1H), 2.62–2.67 (m, 2H), 4.23 (q, $J$ = 7.1 Hz, 2H), 4.49 (dd, $J$ = 8.9, 6.6 Hz,1H), 5.05–5.09 (m, 2H), 5.71–5.78 (m, 1H); $^{13}$C NMR (150 MHz, CDCl$_3$) δ 14.3, 26.8, 28.0, 35.3, 42.3, 57.6, 61.7, 83.8, 117.8, 134.8, 149.5, 171.6, 174.7.

**(2S,4R)-1-tert-butyl 2-ethyl 4-(3-hydroxypropyl)-5-oxopyrrolidine-1,2-dicarboxylate ((2S,4R)-11) and 1-tert-butyl 2-ethyl 5-hydroxy-4-(3-hydroxypropyl)pyrrolidine-1,2-dicarboxylate (12).** To a solution of (2$S$,4$R$)-**10** (100 mg, 0.34 mmol) in THF (3.4 mL), a solution of BH$_3$·THF in THF (1.0 M, 200 μL, 0.20 mmol) was added dropwise at 0˚C under N$_2$. The mixture was stirred at 0˚C for 30 min, then a solution of NaOH (1.0 N, 200 μL, 0.20 mmol) and H$_2$O$_2$ (35%, 170 μL, 1.99 mmol) were added dropwise at 0˚C. After stirring at 0˚C for 30 min, the resulting mixture was quenched with NH$_4$Cl$_{(sat)}$ (2 mL) and diluted with H$_2$O (8 mL). The solvents were evaporated, and the residue was partitioned between H$_2$O (10 mL) and EtOAc (20 mL). The organic layer was washed with H$_2$O (20 mL) and brine (20 mL × 2), dried over MgSO$_4$, filtered and evaporated. The residue was chromatographed (silica gel, 33~50% EtOAc/ $n$-hexane) to provide (2$S$,4$R$)-**11** (23 mg, 0.073 mmol, 21%) and **12** (25 mg, 0.079 mmol, 23%) as colorless oils. (2$S$,4$R$)-**11**: $^1$H NMR (400 MHz, CDCl$_3$) δ 1.30 (t, $J$ = 7.5 Hz, 3H), 1.50 (s, 9H), 1.60–1.67 (m, 2H), 1.91–2.04 (m, 2H), 2.08 (s, 1H), 2.18–2.28 (m, 2H), 2.62–2.70 (m, 1H), 3.65 (t, $J$ = 6.1 Hz, 2H), 4.24 (q, $J$ = 7.1 Hz, 2H), 4.56 (d, $J$ = 9.6 Hz, 1H); $^{13}$C NMR (150 MHz, CDCl$_3$) δ 14.3, 26.7, 28.0, 28.7, 30.0, 41.5, 57.3, 61.8, 62.4, 83.7, 149.6, 171.4, 175.4; ESIMS $m/z$ 338 [M+Na]$^+$; HRESIMS calcd for C$_{15}$H$_{25}$NNaO$_6$ [M+Na]$^+$, 338.1574; found, 338.1586. **12**: $^1$H NMR (400 MHz, CDCl$_3$) δ 1.29 (t, $J$ = 7.1 Hz, 3H), 1.43 (s, 9H), 1.52–1.60 (m, 2H), 1.62–1.67 (m, 2H), 1.89–1.97 (m, 1H), 2.00 (s, 1H), 2.05 (s, 1H), 2.13–2.74 (m, 2H), 3.66 (t, $J$ = 6.1 Hz, 2H), 4.20 (q, $J$ = 7.2 Hz, 2H), 4.26–4.29 (m, 1H), 5.20 (s, 1H); $^{13}$C NMR (100 MHz, CDCl$_3$) δ 14.4, 28.1, 28.3, 28.5, 30.6, 33.3, 58.7, 61.4, 62.4, 81.2, 87.0, 154.4, 173.1; ESIMS $m/z$ 340 [M +Na]$^+$; HRESIMS calcd for C$_{15}$H$_{27}$NNaO$_6$ [M+Na]$^+$, 340.1734; found, 340.1747.

**(2S,4R)-1-tert-butyl 2-ethyl-(3-fluoropropyl)-5-oxopyrrolidine-1,2-dicarboxylate ((2S,4R)-13) and (2S,4S)-1-tert-butyl 2-ethyl-(3-fluoropropyl)-5-oxopyrrolidine-1,2-dicarboxylate ((2S,4S)-13).** Method C: To a solution of DAST (16 μL, 0.12) in CH$_2$Cl$_2$ (0.5 mL), (2$S$,4$R$)-**11** (20 mg, 0.06 mmol) and DIPEA (30 μL, 0.18 mmol) in CH$_2$Cl$_2$ (0.5 mL) was added dropwise at -78˚C under N$_2$. The mixture was stirred at -78˚C for 1 h, and then warmed to room temperature and stirred for another 15 h. The solution was quenched with a mixture of H$_2$O and NaHCO$_{3(sat.)}$ (9/1, 10 mL), acidified to pH = 7~8 with NH$_4$Cl$_{(sat.)}$, and extracted with CH$_2$Cl$_2$ (10 mL × 2). The extracts were washed with brine, dried over MgSO$_4$, filtered and evaporated. The residue was chromatographed (silica gel, 33~50% EtOAc/$n$-hexane) to provide (2$S$,4$R$)-**13** (6 mg, 0.019 mmol, 33%) as a colorless oil.

Method D: To a solution of ($S$)-**8** (200 mg, 0.78 mmol) in THF (4 mL), a solution of lithium bis(trimethylsilyl)amide in THF/ethylbenzene (1.0 M, 0.95 mL, 0.95 mmol) was added dropwise at -78˚C under N$_2$. The mixture was stirred at -78˚C for 1.5 h, then 3-fluoropropyl trifluoromethanesulfonate (328 mg, 1.56 mmol) in THF (2.5 mL) was added dropwise at -78˚C and stirred for 4 h. The resulting mixture was quenched with NH$_4$Cl$_{(sat.)}$ (30 mL), basified to pH = 7~ 8 with NaHCO$_{3(sat.)}$ and extracted with CH$_2$Cl$_2$ (30 mL). The combined extract was washed with brine, dried over MgSO$_4$, filtered and evaporated. The residue was chromatographed (silica gel, 33% EtOAc/$n$-hexane) to provide (2$S$,4$R$)-**13** (32 mg, 0.10 mmol, 13%) and (2$S$,4$S$)-**13** (48 mg, 0.15 mmol, 19%) as colorless oils.

(2S,4R)-**13**: $^1$H NMR (600 MHz, CDCl$_3$) δ 1.30 (t, *J* = 7.1 Hz, 3H), 1.50 (s, 9H), 1.53–1.55 (m, 1H), 1.73–1.82 (m, 2H), 1.95–2.02 (m, 2H), 2.24–2.28 (m, 1H), 2.65–2.68 (m, 1H), 4.24 (q, *J* = 7.1 Hz, 2H), 4.41–4.52 (m, 2H), 4.57 (d, *J* = 9.5 Hz, 1H); $^{13}$C NMR (150 MHz, CDCl$_3$) δ 14.3, 26.5 (d, *J* = 4.4 Hz), 27.9 (d, *J* = 19.8 Hz), 28.0, 28.7, 41.3, 57.2, 61.8, 83.7, 83.8 (d, *J* = 164.4 Hz), 149.6, 171.4, 174.8; ESIMS *m/z* 340 [M+Na]$^+$; HRESIMS calcd for C$_{15}$H$_{24}$FNNaO$_5$ [M+Na]$^+$, 340.1531; found, 340.1545. (2S,4S)-**13**: $^1$H NMR (600 MHz, CDCl$_3$) δ 1.30 (t, *J* = 7.1 Hz, 3H), 1.50 (s, 9H), 1.58–1.63 (m, 1H), 1.66–1.71 (m, 1H), 1.77–1.68 (m, 2H), 1.95–2.01 (m, 1H), 2.53–2.62 (m, 2H), 4.23 (q, *J* = 7.1 Hz, 2H), 4.39–4.53 (m, 3H); $^{13}$C NMR (100 MHz, CDCl$_3$) δ 14.2, 27.3 (d, *J* = 4.7 Hz), 27.9, 28.0 (d, *J* = 21.5 Hz), 28.0, 42.2, 57.6, 61.8, 83.7 (d, *J* = 164.4 Hz), 83.8, 149.5, 171.6, 174.8; ESIMS *m/z* 340 [M+Na]$^+$; HRESIMS calcd for C$_{15}$H$_{24}$FNNaO$_5$ [M+Na]$^+$, 340.1531; found, 340.1536.

**1-(tert-Butyl) 2-ethyl (2R,4S)-4-(3-fluoropropyl)-5-oxopyrrolidine-1,2-dicarboxylate ((2R,4S)-13) and 1-(tert-Butyl) 2-ethyl (2R,4R)-4-(3-fluoropropyl)-5-oxopyrrolidine-1,2-dicarboxylate ((2R,4R)-13).** (2R,4S)-**13** and (2R,4R)-**13** were synthesized according to the procedure for the preparation of (2S,4R)-**13** and (2S,4S)-**13** (Method D) using 1-Boc-D-pyroglutamic acid ethyl ester ((R)-**8**) to afford orange oils (23% and 24%, respectively). (2R,4S)-**13**: $^1$H NMR (600 MHz, methanol-*d$_4$*) δ 1.30 (t, *J* = 7.2 Hz, 3H), 1.50 (s, 9H), 1.67–1.72 (m, 1H), 1.72–1.89 (m, 2H), 1.94–2.04 (m, 2H), 2.21–2.32 (m, 1H), 2.61–2.73 (m, 1H), 4.24 (q, *J* = 7.1 Hz, 2H), 4.40–4.45 (m, 1H), 4.48–4.54 (m, 1H), 4.57 (dd, *J* = 9.4, 0.8 Hz, 1H); $^{13}$C NMR (150 MHz, methanol-*d$_4$*) δ 14.3, 26.5 (d, *J* = 4.8 Hz), 27.9 (d, *J* = 19.8 Hz), 28.0, 28.7, 41.3, 57.2, 61.9, 83.7, 83.8 (d, *J* = 164.3 Hz), 149.6, 171.4, 174.8; ESIHRMS calcd for C$_{15}$H$_{24}$NO$_5$FNa [M+Na]$^+$, 340.1536; found, 340.1537. (2R,4R)-**13**: $^1$H NMR (400 MHz, CDCl$_3$) δ 1.30 (t, *J* = 7.2 Hz, 3H), 1.49 (s, 9H), 1.57–1.65 (m, 1H), 1.65–1.73 (m, 1H), 1.73–1.91 (m, 2H), 1.95–2.02 (m, 1H), 2.52–2.62 (m, 2H), 4.23 (q, *J* = 7.1 Hz, 2H), 4.39–4.47 (m, 1H), 4.48–4.53 (m, 2H); $^{13}$C NMR (100 MHz, CDCl$_3$) δ 14.2, 27.2 (d, *J* = 4.8 Hz), 27.8, 27.90 (d, *J* = 19.1 Hz), 27.91, 42.1, 57.5, 61.7, 83.66 (d, *J* = 164.4 Hz), 83.74, 149.4, 171.5, 174.9; ESIHRMS calcd for C$_{15}$H$_{24}$NO$_5$FNa [M+Na]$^+$, 340.1536; found, 340.1537.

**(2S,4R)-2-(tert-Butoxycarbonylamino)-4-(3-fluoropropyl)pentanedioic acid ((2S,4R)-14).** To a solution of (2S,4R)-**13** (70 mg, 0.22 mmol) in THF (1.7 mL), LiOH$_{(aq)}$ (2.5 N, 1.6 mL, 4.00 mmol) was added and stirred for 2 h at room temperature. The resulting mixture was adjusted to pH = 2 with 1 N HCl and extracted with Et$_2$O (10 mL × 2). The extracts were dried over MgSO$_4$, filtered, and evaporated. The residue was dissolved in Et$_2$O (10 mL) and partition with LiOH$_{(aq)}$ (0.1 N, 10 mL). The aqueous layer was washed with Et$_2$O (10 mL × 2), adjusted to pH = 2 with 1 N HCl and extracted with Et$_2$O (10 mL × 2). The extracts were dried over MgSO$_4$, filtered, and evaporated to provide (2S,4R)-**14** (39 mg, 0.13 mmol, 59%) as a colorless oil. $^1$H NMR (400 MHz, DMSO-*d$_6$*) δ 1.38 (s, 9H), 1.50–1.64 (m, 5H), 1.91–1.97 (m, 1H), 2.39–2.40 (m, 1H), 3.81–3.86 (m, 1H), 4.35–4.49 (m, 2H), 7.13 (d, *J* = 8.1 Hz, 1H), 12.46 (s, 2H); $^{13}$C NMR (100 MHz, DMSO-*d$_6$*) δ 27.8 (d, *J* = 19.3), 28.2 (d, *J* = 4.8 Hz), 28.4, 32.9, 41.3, 52.1, 78.4, 83.8 (d, *J* = 160.7 Hz), 155.9, 174.3, 176.2; ESIMS *m/z* 330 [M+Na]$^+$; HRESIMS calcd for C$_{13}$H$_{22}$FNNaO$_6$ [M+Na]$^+$, 330.1329; found, 330.1325.

**(2R,4S)-2-((tert-Butoxycarbonyl)amino)-4-(3-fluoropropyl)pentanedioic acid ((2R,4S)-14).** (2R,4S)-**13** (160 mg, 0.50 mmol) and LiOH$_{(aq)}$ (2.5 N, 4.1 mL) in THF (3.2 mL) were stirred at room temperature for 1 h. The resulting mixture was acidified with 1 N HCl to pH = 2 and extracted with EtOAc (20 mL). The organic layer was washed with brine, dried over MgSO$_4$, filtered, and evaporated to afford (2R,4S)-**14** (161 mg, 0.50 mmol, quantitative yield) as a light yellow oil. $^1$H NMR (400 MHz, methanol-*d$_4$*) δ 1.22–1.33 (m, 1H), 1.43 (s, 9H), 1.52–1.82 (m, 4H), 2.08–2.27 (m, 1H), 2.44–2.64 (m, 1H), 4.12 (dd, *J* = 10.7, 3.9 Hz, 1H), 4.36 (t, *J* = 5.3 Hz, 1H), 4.48 (t, *J* = 5.3 Hz, 1H); $^{13}$C NMR (100 MHz, methanol-*d$_4$*) δ 28.7, 29.2 (d,

$J$ = 5.2 Hz), 29.6 (d, $J$ = 5.2 Hz), 34.8, 42.8, 53.3, 80.5, 84.4 (d, $J$ = 163.2 Hz), 158.1, 175.9, 178.5; ESIHRMS calcd for $C_{13}H_{22}FNNaO_6$ [M+Na]$^+$, 330.1329; found, 330.1330.

**Di-tert-butyl (2S,4S)-2-((tert-butoxycarbonyl)amino)-4-(3-(((4-nitrophenyl)sulfonyl) oxy)-propyl)pentanedioate ((2S,4S)-15) [6].** (2*S*,4*S*)-**6** (120 mg, 0.29 mmol) was dissolved in $CH_2Cl_2$ (3 mL) and cooled in an ice-bath. After addition of triethylamine (176 mg, 1.74 mmol) and 4-nitrophenylsulfonyl chloride (243 mg, 1.10 mmol), the mixture was stirred for 0.5 h. The solvent was evaporated and the crude product was purified by flash chromatography (10–20% ethyl acetate/*n*-hexane) to afford (2*S*,4*S*)-**15** (145 mg, 0.24 mmol, 84%) as a colorless oil. $^1$H NMR (600 MHz, CDCl$_3$) δ 1.42 (s, 9H), 1.43 (s, 9H), 1.46 (s, 9H), 1.55–1.64 (m, 2H), 1.64–1.91 (m, 4H), 2.25–2.35 (m, 1H), 4.05–4.13 (m, 1H), 4.13–4.19 (m, 2H), 4.87 (d, $J$ = 8.5 Hz, 1H), 8.09–8.15 (m, 2H), 8.39–8.44 (m, 2H); $_{13}$C NMR (100 MHz, CDCl$_3$) δ 26.6, 28.1, 28.2, 28.4, 34.7, 42.8, 53.0, 71.4, 80.0, 81.3, 82.3, 124.6, 129.4, 142.0, 150.9, 155.7, 171.7, 174.4; ESIHRMS calcd for $C_{27}H_{42}N_2O_{11}SNa$ [M+Na]$^+$, 625.2407; found, 625.2425.

## Radiosynthesis of [$^{18}$F]FSPG

[$^{18}$F]FSPG was automatically synthesized in a modified GE TRACERlab FxFDG synthesizer (GE Healthcare, Wauwatosa, WI, USA) as described previously (Supporting information (SI), Fig 45 in S1 File) [6, 21]. [$^{18}$F]Fluoride was produced by irradiating [$^{18}$O]H$_2$O (1.4~2.0 mL) with protons in a silver or niobium target by a GE PETtrace cyclotron. The target entrance window was made of Havar and irradiated for 10–60 min with 40 μA of proton beam. The measured yield of [$^{18}$F]fluoride was approximately 180 mCi/μA for a 30-minute irradiation at the PET center, National Taiwan University Hospital (NTUH). The [$^{18}$F]fluoride/[$^{18}$O]H$_2$O solution was transferred from the target to the nucleophilic synthesizer using inert gas pressure (helium gas, 5N5 grade), [$^{18}$F]fluoride was trapped on a QMA cartridge (Waters, SepPak light) from the [$^{18}$O]H$_2$O, and then [$^{18}$F]fluoride was eluted with 1.5 mL Kryptofix-2.2.2./K$_2$CO$_3$ solution (in H$_2$O/MeCN) into a glass reaction vessel in the synthesizer. The water was evaporated using a stream of helium at 120˚C for 4 minutes. Another 1 mL of MeCN was added followed by azeotropic distillation at 120˚C for 4 minutes to produce the anhydrous [$^{18}$F]KF/ K$_{2.2.2}$ for the following nucleophilic radiofluorination.

Precursor (2*S*,4*S*)-**15** (10 mg) in MeCN (1 mL) was added to the dried residue and heated at 70˚C for 5 minutes. The reaction mixture was cooled to 40˚C, 1 N HCl (4 mL) was added, and then the mixture was heated at 120˚C for 10 minutes to remove the *tert*-Boc protecting groups. After cooling to 40˚C, the mixture was diluted with water (110 mL) and passed through a HR-P cartridge (Chromafix HR-P, Macherey-Nagel). Subsequently, the mixture was *in-line* purified by three MCX cartridges (Oasis MCX, Waters) to remove unreacted [$^{18}$F]fluoride and (2*S*,4*S*)-**15**, and the MCX cartridges were washed with saline (20 mL). The retained [$^{18}$F]FSPG was eluted out from the MCX cartridges with a buffer solution of Na$_2$HPO$_4$·2H$_2$O (70 mg) and NaCl (60 mg) in water (10 mL), followed by passing through an Alumina N cartridge to remove residual [$^{18}$F]fluoride to afford pure [$^{18}$F]FSPG. The resulting [$^{18}$F]FSPG solution was passed through a sterile filter into a sterile vial to provide [$^{18}$F]FSPG injection in an isotonic solution ready for quality control (QC) and clinical use (SI, Fig 46 in S1 File). The detail specifications for [$^{18}$F]FSPG injection is shown in SI, Table 2 in S1 File.

## High performance liquid chromatography analysis

Two high performance liquid chromatography (HPLC) systems: 1) SHIMADZU® LC-6AD equipped with a SPD-20A UV detector; and 2) SHIMADZU® LC-10AT equipped with a SPD-M10A DAD detector (Kyoto, Japan), were used in this study. The columns and mobile phase combinations were used as follows:

1. (2R,4S)-**1**, (2S,4S)-**1**, (2R,4R)-**1**, and (2S,4R)-**1**: a) Phenomenex Luna 5μ C18(2) 100A 250 x 4.60 mm; 2 mM $CuSO_4$; $t_R$ = 4.4 min. b) Phenomenex Chirex 3126 (D)-penicillamine 30 x 4.6 mm; MeCN:2 mM $CuSO_4$ = 15:85; $t_R$ = 7.4, 8.6, 10.1, and 19.4 min. c) Phenomenex Chirex 3126 (D)-penicillamine 30 x 4.6 mm; IPA:2 mM $CuSO_4$ = 15:85; $t_R$ = 4.6, 5.1, 6.4, and 10.1 min (SI, Table 1 in S1 File).

2. [$^{18}$F]FSPG and [$^{18}$F](2R,4R)-**1**: Phenomenex Chirex 3126 (D)-penicillamine 30 x 4.6 mm; IPA/2 mM $CuSO_4$ = 10/90; $t_R$ = 5.3 and 7.4 min (SI, Fig 47 in S1 File).

3. (S)-**3** and (R)-**3**: Chiralcel OD (DAICEL), 250 × 4.6 mm + 50 × 4.6 mm; IPA/*n*-hexane = 1/99; $t_R$ = 10.3 and 7.7 min (SI, Figs 41 and 42 in S1 File).

4. (2S,4S)-**5** and (2R,4R)-**5**: a) Phenomenex Chirex 3126 (D)-penicillamine 30 x 4.6 mm; IPA/2 mM $CuSO_4$ = 15/85; $t_R$ = 8.7 and 12.8 min. b) Chiralcel OD (DAICEL), 250 × 4.6 mm + 50 × 4.6 mm; IPA/*n*-hexane = 1/99; $t_R$ = 6.7 and 5.4 min.

5. (2S,4S)-**6**: Phenomenex Luna 5μ C18(2) 100A 250 x 4.60 mm; MeCN/0.5% TFA = 80/20; $t_R$ = 4.8 min.

6. (2S,4S)-**6** and (2R,4R)-**6**: Chiralcel OD (DAICEL), 250 × 4.6 mm + 50 × 4.6 mm; IPA/*n*-hexane = 2/98; $t_R$ = 22.1 and 20.5 min (SI, Figs 43 and 44 in S1 File).

## Results/Discussion

There are two chiral centers in FSPG, and therefore four FSPG stereoisomers (i.e., (2S,4S)-**1**, (2R,4R)-**1**, (2S,4R)-**1**, and (2R,4S)-**1**). To date, only the (2S,4S)-**1** and (2S,4R)-**1** have been reported [22], whereas (2R,4R)-**1** and (2R,4S)-**1** have not been studied yet.

### Synthesis of (2S,4S)- and (2R,4R)-isomers of FSPG

Preparation of (2S,4S)-**1** and (2R,4R)-**1** is shown in S1 Scheme. An initial attempt to synthesize di-*tert*-butyl protected glutamate (S)-**3** by treatment of *N*-Boc-L-Glu-OH ((S)-**2**) with $(Boc)_2O$ and *t*-butanol in the presence of 4-(dimethylamino)pyridine (DMAP) failed to provide (S)-**3**. Esterification of (S)-**2** with *t*-butanol and dicyclohexylcarbodiimide (DCC) successfully yielded (S)-**3**, but significant racemization was observed. In the beginning, the enantiomeric purity of (S)-**3** was not measured until the target compound (2S,4S)-**1** afforded later was found to be in poor enantiomeric purity. Due to the lipophilic character of compound **3**, the enantiomer excess (ee) was determined by normal-phase chiral HPLC using a Chiralcel OD column and 1% 2-propanol (IPA) in *n*-hexane as mobile phase. The retention times ($t_R$) of (S)-**3** and (R)-**3** were 10.3 and 7.7 min, respectively (SI, Fig 41 in S1 File). *N*-Boc-*L*-glutamic acid 5-*tert*-butyl ester ((S)-**4**) was selected as an alternative starting material for the preparation of (2S,4S)-**1**. Treatment of (S)-**4** with $(Boc)_2O$, *t*-butanol, and DMAP produced (S)-**3** in high yield and optical purity, whereas treatment of (S)-**4** with DCC, *t*-butanol, and DMAP obtained (S)-**3** with a low ee.

The relationship between racemization and the reagents used in the *t*-butyl protection of (S)-**4** was investigated to determine the critical factors for enantiomeric purity of (S)-**3** (Table 1). As shown in Entry 1, using 2.4 eq of DCC, 1.0 eq of DMAP, and 2.2 eq of *t*-BuOH provided (S)-**3** in 44% ee. The same reaction could not proceed without the presence of DMAP (Entry 2), whereas 0.2 eq of DMAP was sufficient for esterification without significant influence over enantiomeric purity (37%, Entry 3). Using 0.94 eq of $(Boc)_2O$ and 0.5 eq of DMAP in an excess amount of *t*-BuOH (i.e., used as solvent) yielded (S)-**3** in 93% ee. When

**Table 1. Enantiomer excess of (S)-3 prepared from different reaction conditions[a].**

| Entry | DCC (eq) | DMAP (eq) | t-BuOH (eq) | (Boc)₂O (eq) | (S)-3 (ee, %) |
|---|---|---|---|---|---|
| 1 | 2.4 | 1.0 | 2.2 | | 44 |
| 2 | 2.4 | 0 | 2.2 | | No reaction |
| 3 | 2.4 | 0.2 | 2.2 | | 37 |
| 4 | | 0.5 | solvent | 0.94 | 93 |
| 5 | | 0.2 | solvent | 0.94 | 93 |
| 6 | | 0.05 | solvent | 0.94 | 95 |

[a] The samples for analysis were collected after 4 h of coupling reaction.

using 0.2 and 0.05 eq of DMAP in the same reaction, the (S)-3 was obtained in high yields with 93 and 95% ee, respectively (Entry 5 and 6; SI, Fig 42 in S1 File). From the above data, DCC was the most critical factor to cause racemization and a catalytic amount of DMAP was sufficient for the t-butyl protection reaction using (Boc)₂O with satisfactory enantiomeric purity.

(S)-3 was converted to (2S,4S)-6 in two steps according to the literature procedures, with modification [6]. Alkylation of the enolate of (S)-3 with various electrophiles only provided the (2S,4S)-isomers; this result, may be due to thermodynamic equilibrium of the enolate intermediates. Therefore, preparation of the (2S,4S)- and (2R,4R)-isomers, starting from (S)-3 and (R)-3, were straightforward, and there was no detectable (2S,4R)- and (2R,4S)-isomers in the reaction mixtures. However, this high stereoselectivity makes the preparation of diastereomeric (2S,4R)- and (2R,4S)-isomers a more challenge task. Treatment of (S)-3 with LiHMDS, followed by allylbromide, selectively formed (2S,4S)-5 in 91% yield. Alcohol (2S,4S)-6 was produced via hydroboration of (2S,4R)-5 and oxidation with H₂O₂ in 85% yield. Fluorination of (2S,4S)-6 using (diethylamino)sulfur trifluoride (DAST) produced (2S,4S)-7 in 49% yield. (2S,4S)-1 was obtained via a simultaneous t-butyl and N-Boc deprotection of (2S,4S)-7 by using trifluoroacetic acid (TFA). The ¹H- and ¹³C-NMR spectra of (2S,4S)-1 were identical with the data previously published [6]. Alternatively, direct 4-alkylation of (S)-3 with 1-bromo-3-fluoropropane successfully provided (2S,4S)-7 in one step with 80% yield. Then, (2R,4R)-1 was synthesized, starting from (R)-4 in three steps by using this more concise synthetic route as shown in S1 Scheme.

## Synthesis of (2S,4R)- and (2R,4S)-isomers of FSPG

Pyroglutamate-derived enolates were reported to react with activated electrophile, such as allyl group, to form a diastereomeric mixture of 4-substituted pyroglutamates with a little stereoselectivity for the trans-isomers [23]. The protected pyroglutamates 8 were chosen as starting materials for the preparation of FSPG isomers (2S,4R)-1 and (2R,4S)-1, as shown in S2 Scheme. Initial treatment of pyroglutamate (S)-8 with LiHMDS, followed by 1-bromo-3-fluoropropane at -78°C, did not form any product. After the reaction mixture was warmed to room temperature, the lithium enolate of (S)-8 reacted with another (S)-8 molecule in situ to form self-addition dimers 9. A similar ring-opening reaction of N-urethane protected pyroglutamates with lithium carbanions was reported [24]. Therefore, (S)-8 was reacted with the more active ally bromide according to the literature method [23], and the 4-alkylation products (2S,4R)-10 and (2S,4S)-10 were afforded in a ratio of 2:1 (37% vs 18%). Hydroboration of (2S,4R)-10, followed by oxidation with H₂O₂, produced alcohol (2S,4R)-11 and an overreduced byproduct 12 in 21 and 23% yields, respectively. Fluorination of (2S,4R)-11 using DAST obtained (2S,4R)-13 in 36% yield. Since the total yield of (2S,4R)-13 starting from

pyroglutamate (*S*)-**8** was only 3% in three-steps, a more reactive alkylation agent, 3-fluoropropyl triflate, was used as an activated electrophile to improve the alkylation yield. Treatment of (*S*)-**8** with LiHMDS, followed by 3-fluoropropyl triflate, afforded a mixture of (2*S*,4*R*)-**13** and (2*S*,4*S*)-**13** in a ratio of 2:3, which was easily separated by flash column chromatography. By using this synthetic route, (2*S*,4*R*)-**13** was directly obtained from (*S*)-**8** by one-step in 13% yield. In a similar manner, (2*R*,4*S*)-**13** and (2*R*,4*R*)-**13** were directly synthesized from (*R*)-**8** and 3-fluoropropyl triflate in 23 and 24% yields, respectively.

Diacid (2*S*,4*R*)-**14** was prepared via hydrolysis of (2*S*,4*R*)-**13** with LiOH, and the *N*-Boc protection of (2*S*,4*R*)-**14** was removed by HCl in EtOAc to obtain (2*S*,4*R*)-**1**, as shown in S2 Scheme. The corresponding enantiomer (2*R*,4*S*)-**14** was prepared via a similar reaction sequence, starting from pyroglutamate (*R*)-**8** in two-steps and a 23% total yield. Then, (2*R*,4*S*)-**14** was deprotected via a TFA-catalyzed hydrolysis to furnish (2*R*,4*S*)-**1**. The byproducts (2*S*,4*S*)-**13** and (2*R*,4*R*)-**13** produced in the alkylation step were hydrolyzed using LiOH, followed by TFA, to obtain (2*S*,4*S*)-**1** and (2*R*,4*R*)-**1**, respectively. Their stereochemistry was confirmed to be identical to the authentic (2*S*,4*S*)-**1** and (2*R*,4*R*)-**1** prepared in S1 Scheme by comparison to their retention times in chiral HPLC. Following the synthetic strategy in S2 Scheme, (2*S*,4*R*)-**1**, (2*S*,4*S*)-**1**, (2*R*,4*S*)-**1**, and (2*R*,4*R*)-**1**, all four FSPG stereoisomers, could be effectively prepared via this three-step synthetic process from (*S*)-**8** and (*R*)-**8**, respectively.

1D and 2D NMR experiments were used for the stereochemical assignments of diastereomeric pyroglutamates **10** and **13**. Selected chemical shift data is listed in Table 2, and the steric relationships between the protons in compounds **10** and **13** were determined by NOESY experiments, as shown in Fig 2. In the *trans* isomers (i.e., (2*S*,4*R*)-**10** and (2*S*,4*R*)-**13**), correlations between the $H_{2\alpha}/H_{3\alpha}$ and $H_{3\beta}/H_{4\beta}$ were observed. In addition, a correlation was found between the $H_{3\alpha}$ and $H_{1'}$ on the propyl side chain of (2*S*,4*R*)-**13**. In the *cis* isomers (i.e., (2*S*,4*S*)-**10** and (2*S*,4*S*)-**13**), the NOE correlations were observed between the $H_{2\alpha}/H_{3\alpha}$ and $H_{3\alpha}/H_{4\alpha}$. Moreover, a correlation was observed between the $H_{3\beta}$ and $H_{1'}$ on the allyl side-chain of (2*S*,4*S*)-**10**. Based on this data, the *trans*/*cis* configuration assignments could be established. Furthermore, the assignments of (2*S*,4*R*)-**10** and (2*S*,4*S*)-**10** in this study are consistent with those in the previous study, in which 1D NOE experiments were used to differentiate the two isomers [23].

From the chemical shift data shown in Table 2, several characteristic patterns are recognized for the 4-substituted pyroglutamate stereoisomers. The chemical shifts of the $H_{2\alpha}$, $H_{3\beta}$, and C-3 for the *trans* isomers are more downfield (i.e., 0.05–0.07 ppm, 0.46–0.58 ppm, and 0.7–1.1 ppm, respectively) than those of *cis* isomers, whereas the chemical shifts of the $H_{3\alpha}$ for the *trans* isomers are more upfield (i.e., 0.47–0.59 ppm) than those of *cis* isomers. The chemical shifts of the $H_{4\beta}$ for the *trans* isomers are more downfield (i.e., 0.08–0.09 ppm) than those of the $H_{4\alpha}$ for the corresponding *cis* isomers. Furthermore, the chemical shift differences between the $H_{3\alpha}$ and $H_{3\beta}$ in the *cis* isomers (i.e., 0.75–0.90 ppm) are more prominent than those in the *trans* isomers (i.e., 0.18–0.27 ppm).

## Scale-up synthesis of optically pure (2*S*,4*S*)-6

Preparation of [$^{18}$F]FSPG is shown in S3 Scheme. The radiolabeling precursor (2*S*,4*S*)-**15** was prepared from (2*S*,4*S*)-**6** according to the literature method [6]. To obtain [$^{18}$F]FSPG in high stereochemical purity, it is essential to determine and control the optical purity of (2*S*,4*S*)-**6**. The ee of (2*S*,4*S*)-**6** was determined by using a Chiralcel OD column, and the retention times for (2*S*,4*S*)-**6** and (2*R*,4*R*)-**6** are 22.1 and 20.5 min, respectively (SI, Figs 43 and 44 in S1 File).

As mentioned above, using (*S*)-**2** as starting material and DCC as a coupling reagent produced (2*S*,4*S*)-**6** with a low ee. In this study, recrystallization was also utilized to provide

**Table 2. The chemical shifts (δ, ppm) of diastereomeric pyroglutamates 10 and 13.**

| compound | H$_{2\alpha}$ | H$_{3\alpha}$ | H$_{3\beta}$ | H$_{4\alpha}$ | H$_{4\beta}$ | C-3 |
|---|---|---|---|---|---|---|
| (2S,4R)-10 | 4.54 | 2.01 | 2.19 | – | 2.74 | 27.9 |
| (2S,4S)-10 | 4.49 | 2.48 | 1.73 | 2.65 | – | 26.8 |
| (2S,4R)-13 | 4.57 | 1.99 | 2.26 | – | 2.67 | 28.7 |
| (2S,4S)-13 | 4.50 | 2.58 | 1.68 | 2.59 | – | 28.0 |

(2S,4S)-6 with >98% ee (Table 3). As shown in Entry 1, (2S,4S)-6 (ee = 42%) was recrystallized in hexanes, and the crystal and mother liquor portions were obtained with 21 and 64% ee, respectively. Using the (2S,4S)-6 recovered from mother liquor for the second recrystallization, the crystal and mother liquor portions were yielded with 40 and 88% ee, respectively. After the third recrystallization cycle, the (2S,4S)-6 collected from mother liquor possessed an ee of 100% with 42% recovery (Entry 1). Treatment of (2S,4S)-6 with EtOAc/hexanes (1/9) furnished (2S,4S)-6 with 66, 92, and 98% ee after the first, second, and third cycles of recrystallization, respectively (Entry 2). By using cyclohexane as a solvent, (2S,4S)-6 with 99% ee was obtained after only one cycle of recrystallization, starting from (2S,4S)-6 with 42% ee (Entry 3). Recrystallization of (2S,4S)-6 (ee = 54%, 1.0 g) using cyclohexane resulted in optically pure (2S,4S)-6 (ee = 100%, 280 mg) after a single recrystallization cycle (Entry 4).

## Analysis of FSPG stereoisomers

Because of the short half-life of $^{18}$F-labeled PET drugs (i.e., 110 min), [$^{18}$F]FSPG needs to be prepared on a daily basis; therefore, fast analytical methods for quality control (QC) are especially essential and must be established to assure the safety and quality of [$^{18}$F]FSPG for clinical use. The activation of [$^{18}$F] fluoride by K$_{2.2.2}$/K$_2$CO$_3$ has been a commonly used method to form the naked and highly nucleophilic fluoride, but the basicity of K$_{2.2.2}$/K$_2$CO$_3$ may

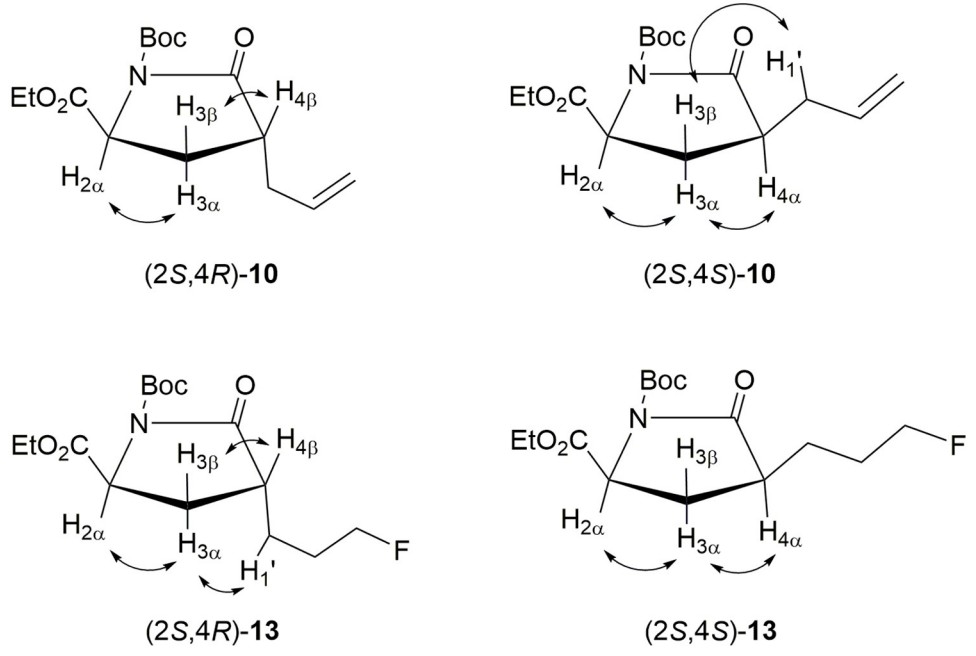

**Fig 2. Stereochemical assignments of pyroglutamates 10 and 13 by NOESY experiments.**

**Table 3. Conditions and results of resolution of (2S,4S)-6 via recrystallization[a,b].**

| | starting material | | | 1st recrystallization | | 2ed recrystallization | | 3rd recrystallization | |
|---|---|---|---|---|---|---|---|---|---|
| Entry | ee (%) | weight (mg) | solvent | crystal (ee, %) | mother liquor (ee; recovery) | crystal (ee, %) | mother liquor (ee; recovery) | crystal (ee, %) | mother liquor (ee; recovery) |
| 1 | 42 | 200 | hexanes | 21 | 64%; 75% | 40 | 88%; 51% | 38 | 100%; 42% |
| 2 | 42 | 200 | EtOAc/ hexanes = 1/9 | 28 | 66%; 91% | 32 | 92%; 41% | 46 | 98%; 36% |
| 3 | 42 | 200 | cyclohexane | 27 | 99%; 24% | | | | |
| 4 | 54 | 1000 | cyclohexane | 37 | 100%; 28% | | | | |

[a] The precipitated crystals possessed lower ee than the starting material, whereas the mother liquors exhibited higher ee than the starting material.

[b] Depicted are the ee values and the recovery of enriched (2S,4S)-6 in the mother liquors that were calculated based on the weight of (2S,4S)-6 used in each recrystallization.

promote the epimerization of the C-2 position of [18F]FSPG [22]. In fact, racemization and epimerization of the intermediates were also observed in other synthetic steps.

Previous HPLC methods for the analysis of FSPG include using a Hypercarb column with a Corona detector and a C18 column with a precolumn derivatization using OPA-reagent and monitored at 340 nm [6]. To the best of our knowledge, these methods were only used to analyze diastereomeric (2S,4S)-1 and (2S,4R)-1 with moderate resolution [22], and no HPLC method has yet been reported for analysis of four FSPG stereoisomers. $CuSO_4$ has been used as an additive in the mobile phase for HPLC analysis, in which amino acid derivatives could be directly monitored by UV detector at 254 nm with sufficient sensitivity [25]. In this study, RP-HPLC analysis of FSPG stereoisomers using a Luna C18 column with a mobile phase of $CuSO_{4(aq)}$ was conducted and monitored at 254 nm. All four stereoisomers displayed the same single peak at 4.4 min, which shows this method is useful for the determination of chemical purity of compound **1** but is not suitable for the determination of enantiomeric or diastereomeric purities of compound **1**. A concise and efficient analytic method for the determination of all four FSPG stereoisomers was pursued.

The Chirex 3126 (d)-penicillamine column has been widely used in the analysis of amino acids, including glutamic acid and aspartic acid. When a solution of 2 mM $CuSO_{4(aq)}$/MeOH (85:15) was used as mobile phase, enantiomers of aspartic acid were separated with high resolution by HPLC. Therefore, an aqueous $CuSO_4$-based mobile phase and Chirex 3126 column were chosen as the starting points for the development of a more convenient and faster HPLC method for analysis of four FSPG stereoisomers without derivatization. Initial analysis using Chirex 3126 (d)-penicillamine column (150 mm × 4.6 mm), with $CuSO_{4(aq)}$ as the mobile phase, and monitored at 254 nm was not satisfactory. Due to the significantly higher lipophilicity of FSPG than that of glutamic acid, a much higher content of organic modifier was needed to elute stereoisomers of **1**. This response therefore significantly increased the back pressure of the chromatographic system. For Chirex 3126 column, the recommended limits for 2-propanol (IPA) and acetonitrile (MeCN) are <15%, and the upper limits of back pressure was 3000 psi. Using Chirex 3126 column (150 mm) with a MeOH/$CuSO_{4(aq)}$ mobile phase (15:85; 27°C), none of FSPG stereoisomers could be eluted out in 60 min (Fig 3a). When IPA/$CuSO_{4(aq)}$ was used as mobile phase (15:85), only (2R,4S)-**1**, (2S,4S)-**1**, and (2R,4R)-**1** were eluted out in 30 min at 25°C (Fig 3b); whereas (2R,4S)-**1**, (2S,4S)-**1**, and (2R,4R)-**1** could be eluted out in 20 min at 35°C (Fig 3c). Under these conditions, the absorption peaks of (2R,4S)-**1** and (2S,4S)-**1** were completely overlapped. By using Chirex 3126 column (150 mm) with a MeCN/$CuSO_{4(aq)}$ mobile phase, the resolution of FSPG stereoisomers was significantly increased. The

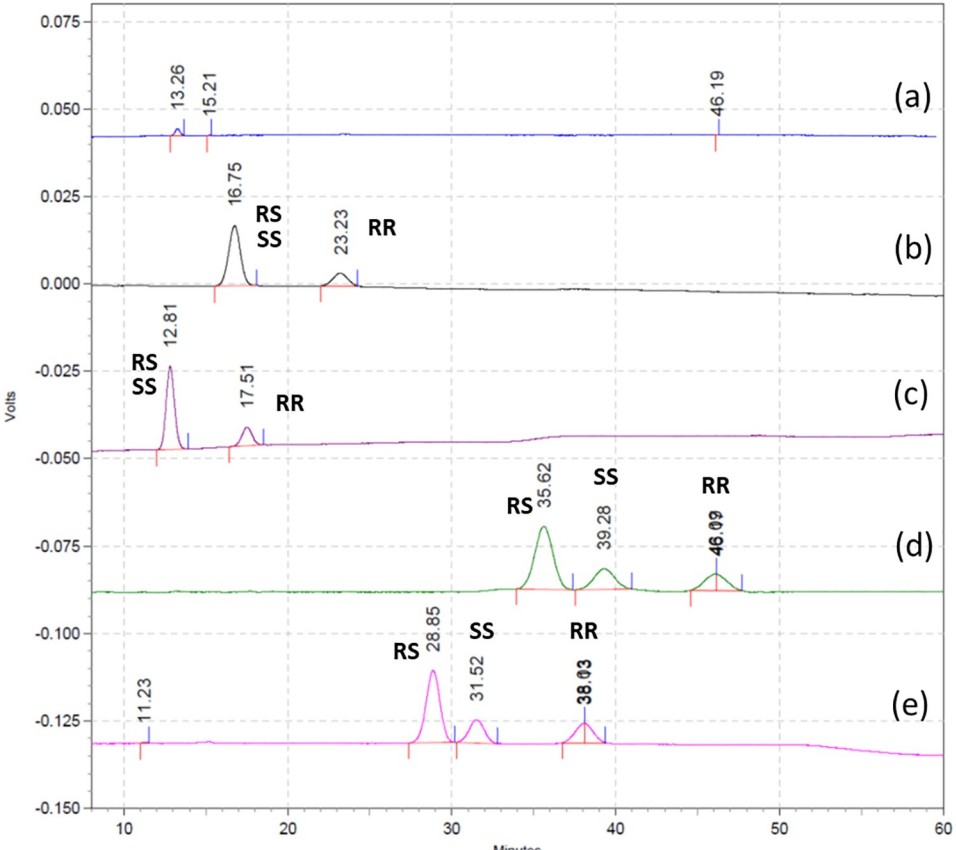

**Fig 3. Chiral HPLC chromatograms.** Using Chirex 3126 column: 150 mm × 4.6 mm. a) MeOH:2 mM $CuSO_4$ = 15:85 at 27 ˚C; b) IPA:2 mM $CuSO_4$ = 15:85 at 25˚C; c) IPA:2 mM $CuSO_4$ = 15:85 at 35˚C; d) MeCN:2 mM $CuSO_4$ = 15:85 at 25˚C; e) MeCN:2 mM $CuSO_4$ = 15:85 at 35˚C.

(2R,4S)-**1**, (2S,4S)-**1**, and (2R,4R)-**1** were eluted out in sequence with $t_R$ of 35.6, 39.3, and 46.1 min at 25 ˚C (Fig 3d), and with $t_R$ of 28.9, 31.5, and 38.0 min at 35 ˚C (Fig 3e), respectively. The $t_R$ and resolution of FSPG stereoisomers were sensitive to the temperature and organic modifiers: lower temperature resulted in longer $t_R$ and higher resolution; MeCN provided higher resolution and longer $t_R$ than IPA.

The (2S,4R)-**1** could not be eluted out under the above analytical conditions using Chirex 3126 column (150 mm). Furthermore, the long $t_R$ seems to be not compatible for analysis of the short half-life [18]F-labeled PET tracers. Therefore, chiral HPLC analysis using shorter Chirex 3126 column was conducted and the results are shown in Fig 4. Using a combination of two Chirex 3126 pre column (30 mm × 4.6 mm × 2) with MeCN/$CuSO_{4(aq)}$ mobile phase successfully separated four FSPG stereoisomers in less than 40 min (25 ˚C; Fig 4a), 35 min (30 ˚C; Fig 4b), and 30 min (30 ˚C; Fig 4c), respectively.

To further shorten the time for analysis, using a single Chirex 3126 pre column (30 mm × 4.6 mm) for separation was then conducted. When using IPA and $CuSO_{4(aq)}$ as mobile phase, the resolution was sufficient to separate enantiomers (2S,4S)-**1** and (2R,4R)-**1** (Fig 5a). The $t_R$ was sensitive to the temperature: $t_R$ for (2S,4S)-**1** and (2R,4R)-**1** were 5.1 and 6.4 min at 23˚C, and 4.1 and 5.1 min at 26˚C, respectively (Fig 5a and 5b). The $t_R$ for the diastereomer (2S,4R)-**1** was 10.1 min at 26˚C (Fig 5c). This mobile phase was satisfactory for the analysis of

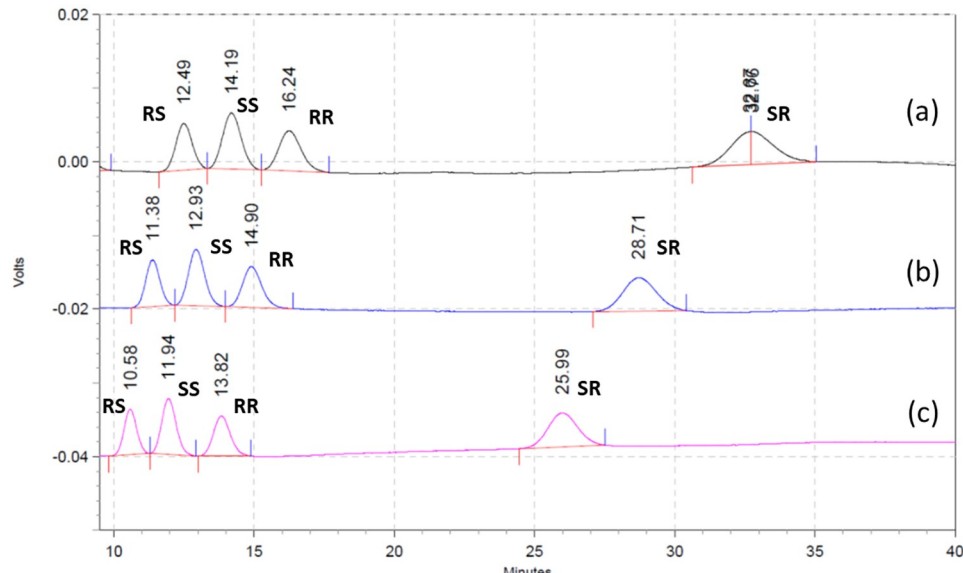

**Fig 4. Chiral HPLC chromatograms.** Using Chirex 3126 column: 60 mm × 4.6 mm; mobile phase: MeCN:2 mM CuSO₄ = 15:85. a) at 25˚C; b) at 30˚C; c) at 35˚C.

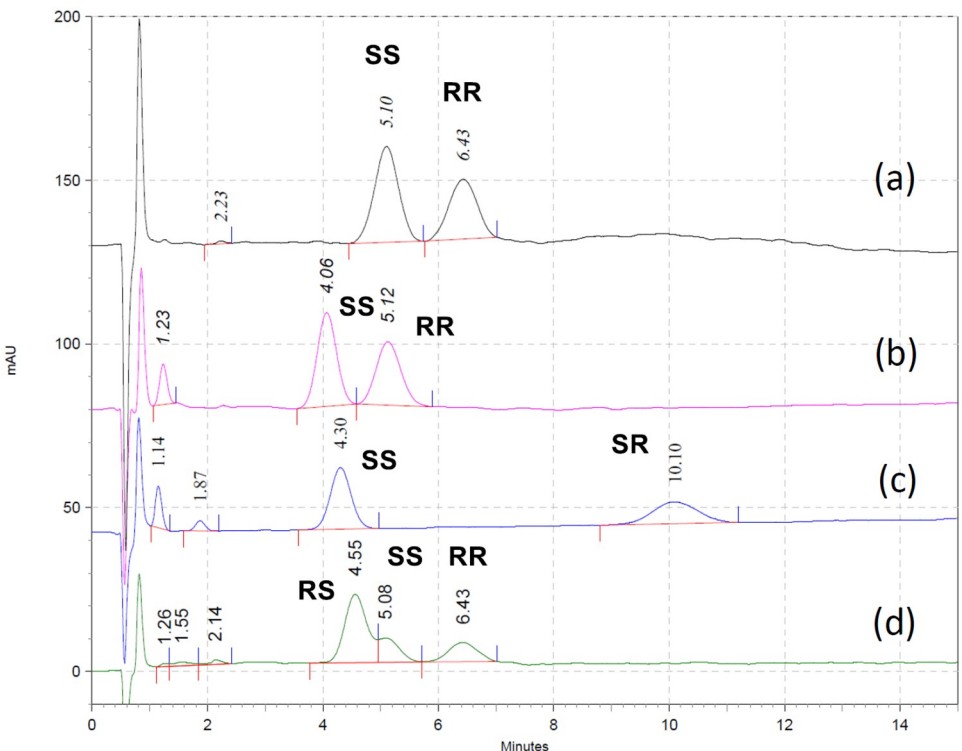

**Fig 5. Chiral HPLC chromatograms.** Using Chirex 3126 column: 30 mm × 4.6 mm; mobile phase: IPA:2 mM CuSO₄ = 15:85. a) (2S,4S)-**1** and (2R,4R)-**1**, at 23˚C; b) (2S,4S)-**1** and (2R,4R)-**1**, at 26˚C; c) (2S,4S)-**1** and (2S,4R)-**1**, at 26˚C; d) (2R,4S)-**1**, (2S,4S)-**1** and (2R,4R)-**1**, at 23˚C.

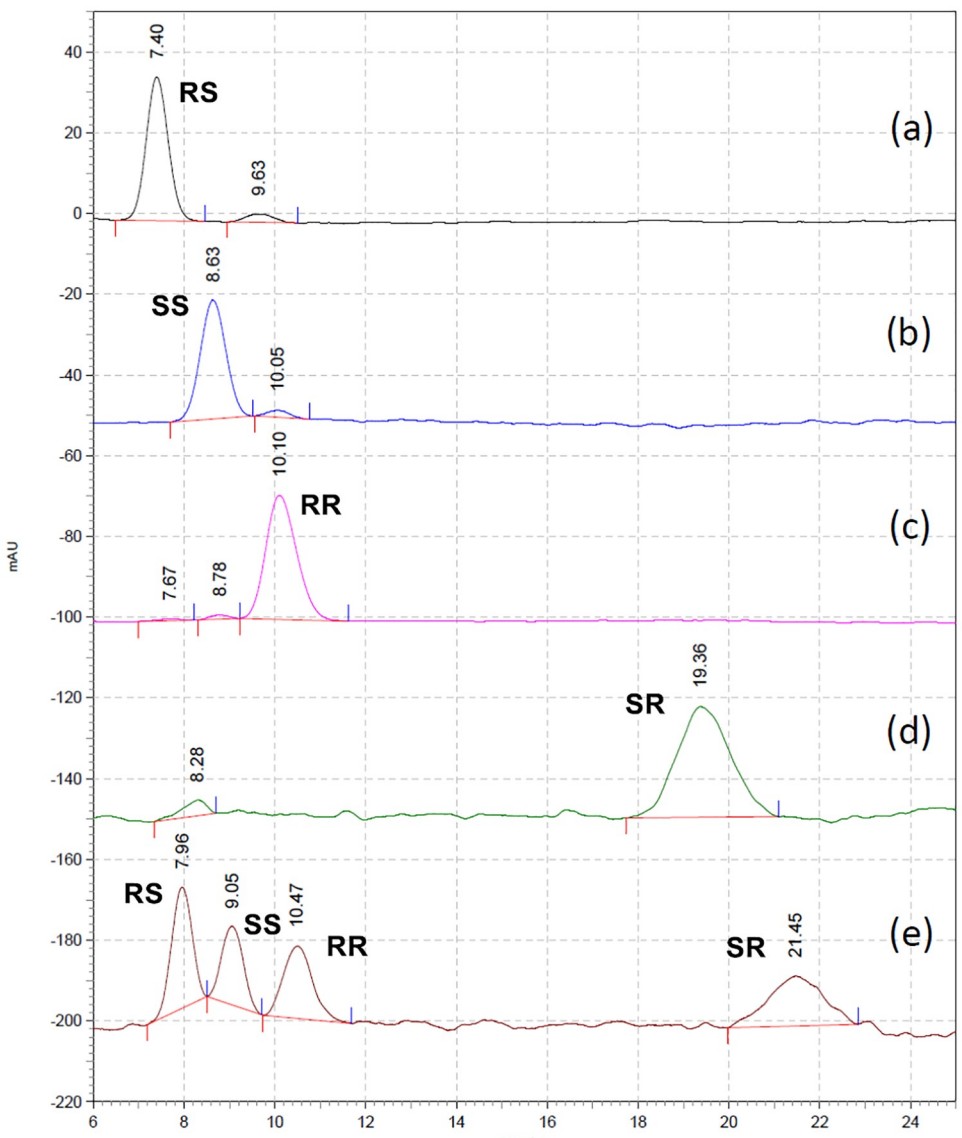

**Fig 6. Chiral HPLC chromatograms.** Using Chirex 3126 column: 30 mm × 4.6 mm; mobile phase: MeCN:2 mM CuSO$_4$ = 15:85; at 22˚C. (a) (2*R*,4*S*)-**1**; (b) (2*S*,4*S*)-**1**; (c) (2*R*,4*R*)-**1**; (d) (2*S*,4*R*)-**1**; (e) a mixture of four FSPG stereoisomers.

(2*S*,4*S*)-**1**, (2*R*,4*R*)-**1**, and (2*S*,4*R*)-**1** in 15 min. The t$_R$ for (2*R*,4*S*)-**1** was 4.6 min at 23˚C, and the absorption peaks of (2*R*,4*S*)-**1** and (2*S*,4*S*)-**1** were partially overlapped (Fig 5d).

Other organic modifiers were used as the mobile phase to improve the resolution between (2*R*,4*S*)-**1** and (2*S*,4*S*)-**1**. Using the short Chirex 3126 pre column (30 mm × 4.6 mm) with a MeCN/CuSO$_{4(aq)}$ mobile phase (15:85; 22˚C) provided satisfactory results for analysis of all four FSPG stereoisomers in 25 min. The individual chromatograms of (2*R*,4*S*)-**1**, (2*S*,4*S*)-**1**, (2*R*,4*R*)-**1**, and (2*S*,4*R*)-**1** are shown in Fig 6a–6d, and their t$_R$ were 7.4, 8.6, 10.1, and 19.4 min, respectively. The chromatogram for a mixture of (2*R*,4*S*)-**1**, (2*S*,4*S*)-**1**, (2*R*,4*R*)-**1**, and (2*S*,4*R*)-**1** is depicted in Fig 6e. In consideration of the different synthetic processes for the preparation of FSPG stereoisomers, (2*R*,4*R*)-**1** could be an impurity in (2*S*,4*S*)-**1** and vice versa (i.e.,

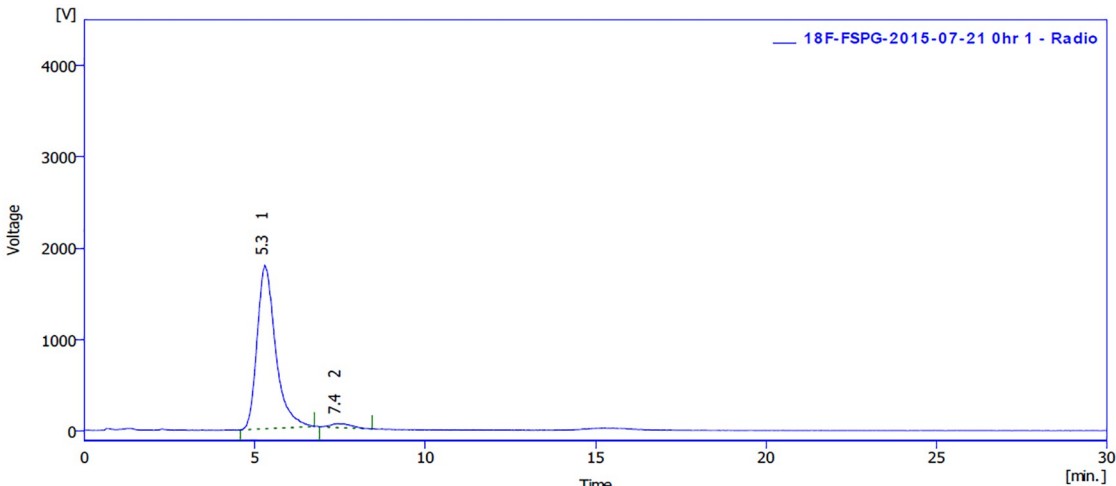

**Fig 7. Chiral HPLC chromatogram of [$^{18}$F]FSPG injection for clinical use.** Using Chirex 3126 column: 30 mm × 4.6 mm; mobile phase: IPA:2 mM CuSO$_4$ = 10:90; at 20˚C.

according to S1 Scheme; Fig 7b and 7c). On the other hand, (2R,4R)-**1** would exist as an impurity in (2R,4S)-**1**, whereas (2S,4S)-**1** was an impurity in (2S,4R)-**1** (i.e., according to S2 Scheme; Fig 7a and 7d). Thus, simultaneous determination of four FSPG stereoisomers has been achieved by using chiral HPLC method, which could be used to ensure the stereoisomeric purity of the [$^{18}$F]FSPG injection for clinical use.

## Radiosynthesis of [$^{18}$F]FSPG

The [$^{18}$F]FSPG injection was prepared in the Radiochemistry Laboratory, PET center, National Taiwan University Hospital (NTUH). All the radiosynthesis and compounding operations for production of the [$^{18}$F]FSPG injection followed the Current Good Manufacturing Practice (cGMP) regulations for human pharmaceuticals standards and is regularly inspected by the Taiwan Food and Drug Administration (TFDA). A detail flow chart of the general procedure for the synthesis of [$^{18}$F]FSPG is shown in SI, Fig 46 in S1 File.

The chiral HPLC chromatogram of [$^{18}$F]FSPG is shown in Fig 7, in which the peak areas for [$^{18}$F]FSPG (t$_R$ = 5.3 min) and [$^{18}$F](2R,4R)-**1** (t$_R$ = 7.4 min) were 97.2% and 2.8%, respectively. In addition, there was no [$^{18}$F](2S,4R)-**1** and [$^{18}$F](2R,4S)-**1** detected in the final [$^{18}$F]FSPG injection. In last few years, [$^{18}$F]FSPG was reliably produced with a radiochemical yield of 4.4±2.4% (EOS) in a synthesis time of 83±9 min from EOB (n = 53). Both the chemical and radiochemical purity of [$^{18}$F]FSPG were >95% with a specific activity of 377±162 mCi/μmol. The injectable dose of [$^{18}$F]FSPG was 8.1±0.3 mCi (300 ± 10 MBq) with a specific activity greater than 95 mCi/μmol at the time of injection, even at 4 hours post synthesis. The amount of nonradioactive FSPG within injected [$^{18}$F]FSPG was not more than 19 μg ($\leq$ 84 nmol).

The detailed specifications for [$^{18}$F]FSPG injection is shown in SI, Table 2 in S1 File. [$^{18}$F]FSPG injection is a clear, colorless, and sterile solution. The radiochemical and enantiomeric purity should be not less than 90%, and the diastereomer contents should be not more than 5%. The limits of residual solvent are $\leq$ 0.5% for EtOH, $\leq$ 0.04% for MeCN, and $\leq$ 0.5% for acetone, respectively. [$^{18}$F]FSPG injection contains residual Kryptofix$_{2.2.2}$ less than 50 μg/mL with a pH range of 5–8. [$^{18}$F]FSPG injection should be stored at controlled room temperature (18~25˚C) with an expiration time of 4 hours.

## Conclusion

To prepare [18F]FSPG with cGMP quality for clinical use, the whole process for production of the [18F]FSPG injection needs to follow official regulations, and obtain permission for human use from the local authority. All four nonradioactive FSPG stereoisomers were prepared as reference standards for development of analytic methods and QC of the final [18F]FSPG injection. An efficient chiral HPLC method capable of simultaneous analysis of four FSPG stereoisomers in 25 min without derivatization through use of a 3-cm chiral column was established. Scale-up synthesis of the key intermediate and precursor for the preparation of [18F] FSPG in high optical purity was achieved via stereo-selective synthesis or resolution by recrystallization. [18F]FSPG has been routinely prepared and used in several PDAC metastasis-related clinical trials at NTUH. Further studies are in progress to evaluate the potential applications of [18F]FSPG for diagnosis of other diseases or disorders.

## Supporting information

**S1 Scheme. Reagents and conditions: a) (Boc)$_2$O, DMAP, *t*-BuOH, rt; b) DCC, DMAP, *t*-BuOH, CH$_2$Cl$_2$, rt; c) lithium bis(trimethylsilyl)amide, THF, -78˚C; d) allyl bromide, THF, -78˚C; e) BH$_3$, THF, 0˚C→rt; f) NaOH, H$_2$O$_2$, 0˚C; g) DAST, DIPEA, CH$_2$Cl$_2$, -78˚C→rt; h) TFA, CH$_2$Cl$_2$, rt; i) 1-bromo-3-fluoropropane, THF, -78˚C.**
(DOCX)

**S2 Scheme. Reagents and conditions: a) lithium bis(trimethylsilyl)amide, THF, -78˚C; b) 1-bromo-3-fluoropropane, THF, -78˚C→rt; c) allyl bromide, THF, -78˚C; d) BH$_3$, THF, 0˚C; e) NaOH, H$_2$O$_2$, 0˚C; f) DAST, DIPEA, CH$_2$Cl$_2$, -78˚C→rt; g) 3-fluoropropyl triflate, THF, -78˚C; h) LiOH, H$_2$O, THF, rt; i) HCl, EtOAc, rt; or TFA, CH$_2$Cl$_2$, rt.**
(DOCX)

**S3 Scheme. Reagents and conditions: a) (4-nitrophenyl)sulfonyl chloride, Et$_3$N, CH$_2$Cl$_2$, rt, 0.5 h; b) [18F]KF, K$_2$CO$_3$, K$_{2.2.2}$, CH$_3$CN, 70˚C, 5 min; c) 1 N HCl, 120˚C, 10 min; d) solid phase extraction: i. H$_2$O, ii. saline, iii. Na$_2$HPO$_4$, NaCl, H$_2$O.**
(DOCX)

**S1 File.**
(PDF)

## Acknowledgments

The authors thanks to Ms Shou-Ling Huang of Ministry of Science and Technology (National Taiwan University) for the assistance in [19]F-NMR experiments.

## Author Contributions

**Conceptualization:** Ling-Wei Hsin.

**Funding acquisition:** Ling-Wei Hsin.

**Investigation:** Kai-Ting Shih, Chia-Ying Yang, Mei-Fang Cheng.

**Resources:** Rouh-Fang Yen.

**Supervision:** Yu-Wen Tien, Chyng-Yann Shiue, Rouh-Fang Yen, Ling-Wei Hsin.

**Writing – original draft:** Ya-Yao Huang.

**Writing – review & editing:** Ling-Wei Hsin.

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
