## [Decision Letter · Decision Letter 0]

6 Oct 2020

PONE-D-20-29935

Synthesis and analysis of 4-(3-fluoropropyl)-glutamic acid stereoisomers to determine the stereochemical purity of (4S)-4-(3-[18F]fluoropropyl)-L-glutamic acid ([18F]FSPG) for clinical use

PLOS ONE

Dear Dr. Hsin,

Thank you for submitting your manuscript to PLOS ONE. After careful consideration, we feel that it has merit but does not fully meet PLOS ONE’s publication criteria as it currently stands. Therefore, we invite you to submit a revised version of the manuscript that addresses the points raised during the review process.

We look forward to receiving your revised manuscript.

Kind regards,

Yu-Hsuan Tsai

Academic Editor

PLOS ONE

Journal Requirements:

2. We note that this submission includes NMR spectroscopy data. We would recommend that you include the following information in your methods section or as Supporting Information files:

1) The make/source of the NMR instrument used in your study, as well as the magnetic field strength. For each individual experiment, please also list: the nucleus being measured; the sample concentration; the solvent in which the sample is dissolved and if solvent signal suppression was used; the reference standard and the temperature.

2) A list of the chemical shifts for all compounds characterised by NMR spectroscopy, specifying, where relevant: the chemical shift (δ), the multiplicity and the coupling constants (in Hz), for the appropriate nuclei used for assignment.

3)The full integrated NMR spectrum, clearly labelled with the compound name and chemical structure.

We also strongly encourage authors to provide primary NMR data files, in particular for new compounds which have not been characterised in the existing literature. Authors should provide the acquisition data, FID files and processing parameters for each experiment, clearly labelled with the compound name and identifier, as well as a structure file for each provided dataset. See our list of recommended repositories here: https://journals.plos.org/plosone/s/recommended-repositories

Additional Editor Comments (if provided):

Please address all points of the reviewers.

Reviewers' comments:

Reviewer's Responses to Questions

**Comments to the Author**

1. Is the manuscript technically sound, and do the data support the conclusions?

Reviewer #1: Yes

Reviewer #2: Yes

2. Has the statistical analysis been performed appropriately and rigorously? 

Reviewer #1: Yes

Reviewer #2: Yes

3. Have the authors made all data underlying the findings in their manuscript fully available?

Reviewer #1: Yes

Reviewer #2: Yes

4. Is the manuscript presented in an intelligible fashion and written in standard English?

Reviewer #1: Yes

Reviewer #2: Yes

5. Review Comments to the Author

Reviewer #1: 1: The half life of 18F is 109.7 min, it is more approprite to round its half life to 110 min, rather than109 min.

2: The separation resolution of four stereoisomers showed in Fig 7 is much better than in Fig 6.

Their corresponding HPLC separation conditions, ranging from column length, solvent (IPA vs MeCN), temperature (23 centigrade vs 22 centigrade) to running time (15 min vs 25 min) are differnt. Compared with Fig 6, which one (or several) is the key point for good resolution? In Fig 6, column temperature proves to be crucial, but in Fig 7, it is hard to draw any conclusions for the improved resolution performance. Studies in Fig 7 will be more helpful only if it could offer some useful advices for people who may use your HPLC condition for similar seperation, otherwise, it is confusing

3: More comparision studies need to be done to figure out what really matters for a good resolution. Temperature? Column length? Solvent or running time? The lack of such experiments devalues this paper.

Reviewer #2: The manuscript by Ling-Wei Hsin et al. submitted to Plos one described Synthesis and analysis of 4-(3-fluoropropyl)-glutamic acid stereoisomers to determine the stereochemical purity of (4S)-4-(3-[18F]fluoropropyl)-L-glutamic acid ([18F]FSPG). This study described in detail the synthesis of the four isomers of 4-(3-Fluoropropyl)-glutamic acid and their HPLC liquid phase conditions, as well as the radiolabeling of [18F]FSPG, which is of great value for clinically promoting the transformation of [18F]FSPG . But there are also the following issues that need to be further supplemented.

1: The F-19 NMR spectra and optical rotations of compounds (2S, 4S)1, (2S, 4R)1, (2R, 4R)1 and (2R, 4R)1 should be added in this manuscript.

2：Why (S) 8 can be converted to (2S,4R) 11, but the related literature of FSPG can only get (2S,4S) derivatives, this reason should be explained in this manuscript.

3: Since the purpose of this article is to promote the clinical production of [18F]FSPG, the detailed steps of radiolabeling in the article should be described, and the average yield of multiple high-dose labeling should be provided.

4：The physical and chemical properties of the final product [18F]FSPG should be provided.

---

## [Author Response · Author response to Decision Letter 0]

6 Nov 2020

This manuscript has been carefully revised according to the comments from the editor and reviewers. A “Point-by-point response to the comments” file is enclosed to outline the changes we made and explanations to the reviewer’ concerns point by point.

---

## [Editor Report · Decision Letter 1]

9 Nov 2020

PONE-D-20-29935R1

Synthesis and analysis of 4-(3-fluoropropyl)-glutamic acid stereoisomers to determine the stereochemical purity of (4S)-4-(3-[18F]fluoropropyl)-L-glutamic acid ([18F]FSPG) for clinical use

PLOS ONE

Dear Dr. Hsin,

Thank you for submitting your manuscript to PLOS ONE. After careful consideration, we feel that it has merit but does not fully meet PLOS ONE’s publication criteria as it currently stands. Therefore, we invite you to submit a revised version of the manuscript that addresses the points raised during the review process.

We look forward to receiving your revised manuscript.

Kind regards,

Yu-Hsuan Tsai

Academic Editor

PLOS ONE

Additional Editor Comments (if provided):

Thanks for submitting the revised manuscript, which has addressed the comments raised by the reviewers. However, there are three points I would like the authors to address.

1. Please add a statement about the use of CuSO4 at 2 mM. Why specifically at this concentration?

2. Most peaks in Figs S13, S15, S17 are difficult to observe, please provide an enlarge image to show those peaks with clear splitting patterns.

3. 13C of (2S,4R)-11 contains too much impurities. Please update the spectrum using pure material.

---

## [Author Response · Author response to Decision Letter 1]

26 Nov 2020

A Point-by-point Response to the Comments file is attached.

---

## [Editor Report · Decision Letter 2]

27 Nov 2020

Synthesis and analysis of 4-(3-fluoropropyl)-glutamic acid stereoisomers to determine the stereochemical purity of (4S)-4-(3-[18F]fluoropropyl)-L-glutamic acid ([18F]FSPG) for clinical use

PONE-D-20-29935R2

Dear Dr. Hsin,

We’re pleased to inform you that your manuscript has been judged scientifically suitable for publication and will be formally accepted for publication once it meets all outstanding technical requirements.

Kind regards,

Yu-Hsuan Tsai

Academic Editor

PLOS ONE

---

## [Editor Report · Acceptance letter]

3 Dec 2020

PONE-D-20-29935R2 

Synthesis and analysis of 4-(3-fluoropropyl)-glutamic acid stereoisomers to determine the stereochemical purity of (4S)-4-(3-[*18*F]fluoropropyl)-L-glutamic acid ([*18*F]FSPG) for clinical use 

Dear Dr. Hsin:

I'm pleased to inform you that your manuscript has been deemed suitable for publication in PLOS ONE. Congratulations! Your manuscript is now with our production department. 

Kind regards, 

on behalf of

Dr. Yu-Hsuan Tsai 

Academic Editor

PLOS ONE